# The non-catalytic role of DNA polymerase epsilon in replication initiation in human cells

Sameera Vipat [1,3], Dipika Gupta [2,3], Sagun Jonchhe[2,3], Hele Anderspuk[1], Eli Rothenberg[2] & Tatiana N. Moiseeva [1] ✉

DNA polymerase epsilon (PolE) in an enzyme essential for DNA replication. Deficiencies and mutations in PolE cause severe developmental abnormalities and cancers. Paradoxically, the catalytic domain of yeast PolE catalytic subunit is dispensable for survival, and its non-catalytic essential function is linked with replicative helicase (CMG) assembly. Less is known about the PolE role in replication initiation in human cells. Here we use an auxin-inducible degron system to study the effect of POLE1 depletion on replication initiation in U2OS cells. POLE1-depleted cells were able to assemble CMG helicase and initiate DNA synthesis that failed shortly after. Expression of POLE1 non-catalytic domain rescued this defect resulting in slow, but continuous DNA synthesis. We propose a model where in human U2OS cells POLE1/POLE2 are dispensable for CMG assembly, but essential during later steps of replication initiation. Our study provides some insights into the role of PolE in replication initiation in human cells.

The core mechanisms of the initiation of DNA replication are conserved from yeast to mammals. Heterohexameric MCM helicase is a central element of the replication complex. MCM is loaded on the chromatin during origin licensing in G1 phase of the cell cycle and serves as a platform for recruiting all other replication components. MCM is activated by its binding partners CDC45 and GINS (forming the CMG helicase) and DDK- and CDK-dependent phosphorylations, while CTF4 brings together the CMG helicase and polymerase alpha/primase complex[1]. After CMG activation, two heterohexameric MCM complexes start moving toward each other, bypassing each other[2–4]. After priming by DNA polymerase alpha (POLA), the initial DNA synthesis during CMG bypass is performed by DNA polymerase delta (POLD), with DNA polymerase epsilon (POLE) taking over the leading strand synthesis after the initial bypass of polymerases past one another[2,5]. After this "polymerase switch" step, two replication forks, each with its own leading and lagging strands, are established.

DNA polymerase epsilon (POLE) is a major DNA polymerase, responsible for synthesizing the leading strand during DNA replication[6]. In yeast, polymerase epsilon (PolE) is recruited to MCM helicase together with GINS complex, playing a critical role in replisome assembly and activation[7–9]. Surprisingly, the N-terminal catalytic domain of PolE catalytic subunit (Pol2 in yeast, POLE1 in humans) is dispensable for cell survival in yeast[10,11]. Since this phenomenon was first observed[10,11], many studies have been conducted to elucidate the non-catalytic role of Pol2 in DNA synthesis and the mechanism allowing cells to survive in the absence of the catalytic domain of PolE[9,12–15]. In the absence of the catalytic N-terminal domain, the C-terminal domain of Pol2 is necessary and sufficient to assemble a full helicase and support replication in vitro[16]. According to structural and biochemical data[13,17], DNA synthesis of both leading and lagging strands in the absence of Pol2 catalytic domain is performed by POLD. In this case, DNA synthesis is slower due to slower DNA unwinding by the CMG helicase[17], suggesting suboptimal CMG activation in these cells. Systems that lack the catalytic domain of Pol2 have proven to be invaluable tools to study the mechanism of replication initiation in eukaryotes.

[1]Department of Chemistry and Biotechnology, Tallinn University of Technology, Tallinn 12618, Estonia. [2]Department of Biochemistry and Molecular Pharmacology, New York University School of Medicine, New York, NY 10016, USA. [3]These authors contributed equally: Sameera Vipat, Dipika Gupta, Sagun Jonchhe. ✉e-mail: Tatiana.Moiseeva@taltech.ee

To our knowledge, no studies have been performed to clarify whether the non-catalytic role of DNA polymerase epsilon is conserved in human cells. In this study, we aimed to identify the essential domains of POLE1, test whether the non-catalytic C-terminal domain of human POLE is sufficient for DNA synthesis, and establish the role of DNA polymerase epsilon in the initiation of DNA replication in human cells.

We have created a cell line in which POLE1 is tagged with mAID (auxin-inducible degron), allowing rapid and efficient depletion of this protein in human U2OS cells without causing a G1 arrest. Surprisingly, POLE1 depletion did not prevent CMG assembly or MCM phosphorylation during ATR inhibition-induced origin firing, indicating that replisome assembly in POLE-deficient cells proceeds to a late stage before DNA replication fails. Indeed, we were able to observe some residual EdU incorporation that was sensitive to aphidicolin, in POLE1-depleted cells.

Using the POLE1-mAID cell line, we show that the C-terminal non-catalytic domain of POLE1 can support DNA replication in human cells in the absence of a full-length protein, although DNA synthesis in such cells is slower. We propose that polymerase delta substitutes for polymerase epsilon in cells expressing the POLE1 C-terminal domain, but is unable to support DNA synthesis in the absence of POLE1/POLE2 due to a failure of the replication initiation, probably at the step of polymerase switching.

POLE insufficiency is known to cause developmental abnormalities in humans[18–20], and drive replication stress and genomic instability in mice and *C. elegans*[21,22], which is generally attributed to insufficient origin firing. However, no specific molecular mechanism has been established to date. Our study provides mechanistic insight into the effects of POLE1 depletion and the role of POLE in DNA replication initiation in human cells.

## Results
### Development and characterization of mAID-tagged POLE1 in U2OS cells
We used an auxin-inducible degron system (mAID) to be able to study the role of DNA polymerase epsilon in replication initiation in human cells. This system is derived from plants and allows rapid depletion of the mAID-tagged protein after it is efficiently ubiquitylated by E3 ligase recruited by F-box protein osTIR1 only in presence of the plant hormone auxin (in this study we used indole-3-acetic acid or 3-IAA)[23]. Following the approach described by Natsume et al.[24], we used CRISPR/Cas9 to knock-in (KI) osTIR1 under a doxycycline-inducible promoter into the AAVS1 "safe harbor" locus, and added a mAID-mCherry tag at the C-terminus of endogenous POLE1 using either one HR template (containing hygromycin resistance gene) or two HR templates (one containing neomycin resistance marker and the other with the hygromycin resistance marker—to increase the probability of homozygous knock-ins). After single-cell cloning, the clonal lines were tested by PCR to ensure homozygous KI of the mAID-mCherry tag (Supplementary Fig. 1a, b). Several homozygous KI clones were identified. Two of the homozygous KI clones—clones 16 (resistant to both neomycin and hygromycin) and 1.6 (sensitive to neomycin)—were treated with 2 μg/ml doxycycline (dox) to induce osTIR1 expression and 500 μM auxin (aux) to promote POLE1 degradation. 24 h treatment with doxycycline alone slightly reduced the level of POLE1 (Fig. 1a). POLE1 level was greatly decreased after the 24 h incubation with both chemicals, confirming the efficiency of the system (Fig. 1a). mAID-mCherry tagging also notably changed the electrophoretic mobility of the POLE1 protein, additionally confirming the absence of the endogenous untagged POLE1 in the tested clones.

In order to test the effect of POLE1 depletion on cell growth we seeded equal numbers of clone 16, clone 1.6, or U2OS cells and treated them with DMSO or dox/aux for 72 h, and counted the cells every 24 h (Fig. 1b). Untreated knock-in cells grew slightly slower compared to the wild-type U2OS cells, however, the addition of dox/aux completely stopped proliferation of clones 16 and 1.6, in agreement with the essential role of POLE1 protein in the cell cycle. Dox/aux treatment also slightly decreased the growth of U2OS cells, as has been previously observed[25].

The effect of short-term POLE1 depletion on replication and cell cycle was evaluated by treating cells with doxycycline for 16 h to induce osTIR1 expression, followed by auxin treatment for 1–8 h. Western blot analysis showed that 1 h of auxin treatment was enough to greatly decrease the level of POLE1 in clone 16 cells (Fig. 1c). EdU incorporation analyses showed that while cells slowed down replication within 1 h of auxin addition, the most dramatic effect was seen after 3 h of auxin treatment, resulting in a strong block of EdU incorporation by S-phase cells (Supplementary Fig. 1c, gating strategy shown on Supplementary Fig. 1o). 16 or 24 h treatment with doxycycline alone (required for osTIR1 induction) visibly reduced levels of POLE1, resulting in EdU incorporation decrease (Supplementary Fig. 1d) and possibly creating stress even before the auxin addition. Therefore, to avoid potential difficulties interpreting results from such experiments, we decided to adhere to 16–24 h concurrent treatments with dox/aux in this study.

In order to ensure that the C-terminal tagging of POLE1 with mAID-mCherry did not have any significant effects of replication dynamics in U2OS cells, we performed a series of experiments comparing U2OS, clone 16, and clone 1.6 (Supplementary Fig. 1e–i). Our data show that POLE1 C-terminal tagging did not significantly affect POLE1 levels (Supplementary Fig. 1e), overall EdU incorporation (Supplementary Fig. 1f), the number of cells in S-phase (Supplementary Fig. 1g), or replication fork speed in U2OS cells (Supplementary Fig. 1h, i).

In order to evaluate the effect of POLE1 depletion on replication and cell cycle, we treated wild-type U2OS cells, clone 16, or clone 1.6 cells with dox/aux for 24 h, followed by a 30 min EdU pulse. Flow cytometry analysis of EdU incorporation and DNA content showed a dramatic decrease in EdU-positive cells after POLE1 depletion (Fig. 1d, e, gating is shown on Supplementary Fig. 1j). Similar results were observed with clone 1.6 (Supplementary Fig. 1k). Surprisingly, there was no notable G1 arrest after 24 h of treatment (Fig. 1f), indicating that POLE1 depletion did not prevent cells from entering S-phase.

### Cell line instability is caused by TetR promoter methylation
In EdU incorporation FACS data, we noticed a subpopulation of POLE1-mAID cells that are resistant to dox/aux treatment and keep EdU incorporation at near-normal level. While the experiments shown above are done using early passages of clone 16, after a few weeks in culture this subpopulation grew, and after 2 months in culture (2 m), nearly 100% of cells did not respond to treatment (Supplementary Fig. 1l). Similarly, treating clone 16 with dox/aux for 10 days selected a population (10d) that was fully resistant to dox/aux treatment (Supplementary Fig. 1m). In order to check that these resistant cells still express mAID-tagged POLE1 and osTIR1, we treated cells with DMSO or dox/aux for 16 h, followed by cell lysis and western blot analysis (Supplementary Fig. 1m). 2 m and 10d cells still expressed mAID-mCherry-tagged POLE1, however, they failed to express osTIR1, which explained failure to deplete POLE1.

Inactivation of the doxycycline-inducible transgenes is a known problem that has been attributed to the methylation of the tetR promoter[26]. In order to confirm that methylation is the reason for doxycycline resistance in our clones, we treated cells with demethylating agent 5-azacytidine (5AC) for 48 h before adding dox/aux for 16 h, and tested POLE1 and osTIR1 levels by western blot. Our data showed that demethylation restored osTIR1 expression and POLE1 degradation in response to dox/aux, thus confirming promoter methylation as the primary reason for the acquired doxycycline resistance (Supplementary Fig. 1n). Although we did not see an increase of Chk1 phosphorylation in our experiment, 5AC is a known

DNA damaging agent[27], and we therefore chose not to use it in the DNA replication experiments. For the data in this paper, we used early passages of the clones 16 and 1.6, disregarding the resistant population when possible. In the experiments where it was not possible to disregard the resistant population (i.e., western blots), the low POLE1 signal seen in dox/aux treated samples is likely coming from the wild-type levels of POLE1 in the resistant population (-10–20%) and does not indicate insufficient depletion of POLE1 in the vast majority of the cells.

### ATRi-induced replication initiation in POLE1-depleted cells

ATR inhibitors (ATRi) have been shown to induce massive origin firing within minutes of treatment resulting in replication proteins' recruitment into the nuclease insoluble chromatin fraction, MCM4 hyperphosphorylation and an increase in EdU incorporation by replicating cells[28,29]. In order to check if ATRi-induced replication initiation is blocked by POLE1 depletion, we treated clone 16 or clone 1.6 cells with DMSO or dox/aux combination for 16 h, added DMSO or 5 μM ATRi (AZD6738) for the last 1 h, and isolated nuclease-insoluble chromatin

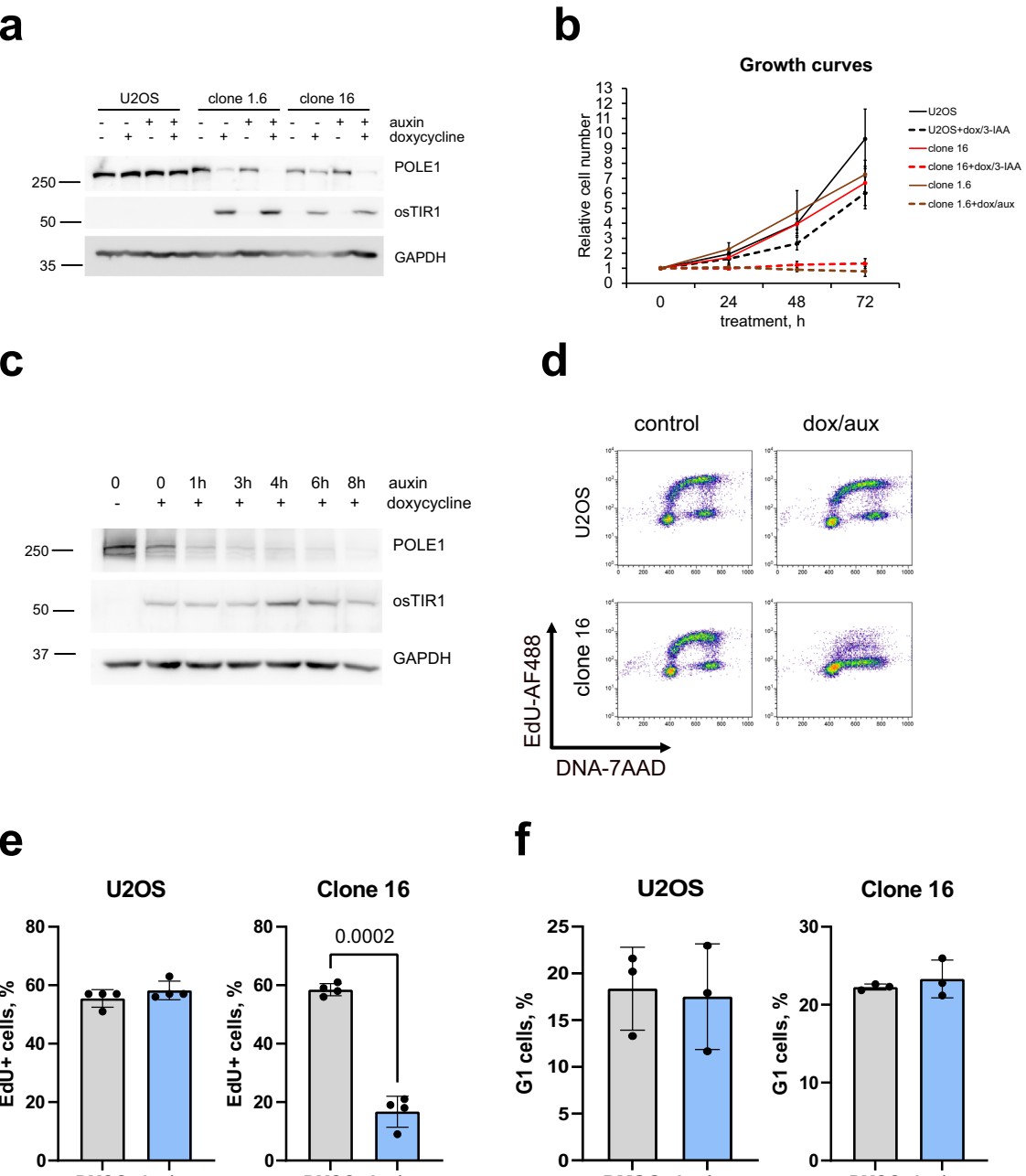

**Fig. 1 | Creating and characterizing a cell line expressing mAID-tagged polE1.**
**a** Wild-type U2OS, homozygous mAID-KI clones 16 and 1.6 were treated for 24 h with doxycycline, auxin, or both, as indicated, western blots of total cell lysates are shown. **b** Equal numbers or wild-type U2OS, or homozygous mAID-KI clones 16 or 1.6, were seeded on 60 mm dishes and treated with DMSO or dox/aux for 72 h. The data are depicted as mean + SD from *n* = 3 independent experiments. **c** Clone 16 was treated with doxycycline overnight (16 h), auxin was added for the indicated times; western blots of total cell lysates are shown. **d–f** Wild-type U2OS, homozygous

mAID-KI clone 16 were treated for 24 h with DMSO or dox/aux, 10 μM EdU was added for the last 30 min of treatment. **d** Flow cytometry plots showing EdU incorporation and DNA content (7-AAD staining) are shown. **e, f** Quantification of the flow cytometry data is shown—mean + SD from n = 3 independent experiments. Paired t-test was used for statistical analyses, p values are shown where they are statistically significant. **e** EdU+ cells were quantified based on EdU signal being above G1 and G2 levels. **f** Cells were assigned to G1 based on DNA content of 2n and being EdU negative. Source data are provided as a Source data file.

fraction (Fig. 2a, b, Supplementary Fig. 2a–d). Our experiment showed that while dox/aux treatment strongly depleted POLE1 both on chromatin and in the soluble protein lysate, it did not prevent MCM4 hyperphosphorylation or CDC45 recruitment to chromatin. In yeast the role of polymerase epsilon in CMG assembly is thought to be in the recruitment of GINS[8]. Therefore, we also studied the recruitment of SLD5 subunit of GINS to the nuclease insoluble chromatin in POLE1-depleted cells in response to ATR. Chromatin SLD5 levels, which increased with ATRi-induced origin firing, were not suppressed by POLE1 depletion (Fig. 2a, b, Supplementary Fig. 2a–d), suggesting that GINS can be recruited to the sites of replication initiation in response to ATRi the absence of POLE1 in human cells.

To further investigate CMG helicase assembly in POLE1-depleted cells, we synchronized U2OS, clone 16, and clone 1.6 cells using thymidine-nocodazole block, and added dox/aux to all the cells during the last 8 h of nocodazole treatment (Fig. 2c). Cells were collected 0 (mitosis), 3, 9, and 12 h following the release from nocodazole into dox/aux containing medium, and the levels of MCM, CDC45, and SLD5 on chromatin were analyzed. All three cell lines have mostly completed mitosis by 3 h, and U2OS cells were in S-phase during the 9 h and 12 h timepoints (Supplementary Fig. 2e). POLE1-depleted clones 16 and 1.6, as expected, could not significantly progress through S-phase due to the lack of efficient DNA synthesis, however, these cells were capable of loading MCM on chromatin in G1 and recruiting CDC45 and GINS to chromatin 9–12 h after the release from nocodazole (Fig. 2c, Supplementary Fig. 2f–h). These data support our hypothesis that POLE1 is not required for CMG assembly in U2OS cells.

In order to test if the ATRi-induced replisome assembly in the absence of POLE1 resulted in increased EdU incorporation, we added ATRi to the clone 16 or clone 1.6 cells after POLE1 depletion, followed by EdU-labeling of ongoing replication. Flow cytometry analysis showed a significant increase in EdU incorporation by S-phase cells even after POLE1 depletion (Fig. 2d, e, Supplementary Fig. 2i). These data indicate that ATR signaling actively suppressed DNA synthesis in POLE1-depleted cells, probably through the same mechanism as during normal, POLE-proficient replication[28].

Western blot with an antibody against CHK1 phosphorylated on serine-345, the canonical ATR-dependent phosphorylation site, confirmed that POLE1 depletion causes ATR activation, probably due to replication stress (Fig. 2a, Supplementary Fig. 2b–d). Accordingly, we observed an increase in ssDNA in POLE1-depleted cells by immunofluorescent detection of CldU under native conditions (Fig. 2f, Supplementary Fig. 2j).

## Aberrant DNA synthesis in POLE1-depleted cells

In the ATRi EdU incorporation experiments (Fig. 2d), we observed a shift in EdU incorporation for the subpopulation of cells that we originally considered EdU negative. In order to verify that these seemingly EdU-negative cells after POLE1 depletion, in fact, do incorporate EdU, just at a slower rate, we depleted POLE1 in clone 16 or clone 1.6 by 16 h dox/aux treatment followed by incubations with EdU for 1–7 h instead of 30 min used in the standard setup (Fig. 3a–c). This experiment confirmed a slow EdU incorporation by POLE1-depleted cells, and allowed us to estimate that POLE1 depletion slows down EdU incorporation ~18–21 times, compared to wild-type levels of POLE1. Slow DNA synthesis by POLE1-depleted cells was sensitive to aphidicolin (Fig. 3d, Supplementary Fig. 3a), implying that a B-family DNA polymerase was responsible for this slow DNA synthesis.

Next, to test whether the cells could complete replication without POLE1, we treated control and POLE1-depleted cells with nocodazole—the inhibitor of microtubule polymerization—for 8 h to induce G2/M arrest (Supplementary Fig. 3b): while in the control sample the majority of the cells accumulated in G2 after nocodazole treatment, POLE1-depleted cells were barely affected by the nocodazole, indicating that these cells did not enter G2 during the time of treatment.

One possible explanation of such slow DNA synthesis is a low number of replication forks, limited by the availability of POLE1. Therefore, we assessed the effect of POLE1 depletion on replication dynamics by performing DNA fiber analysis on vehicle- or dox/aux-treated clone 16 or clone 1.6 cells. POLE1-depleted cells demonstrated "dotty" DNA fibers with an overall lower level of nucleotide incorporation (Fig. 3e, Supplementary Fig. 3c). Some "normal"-looking fibers, observed in dox/aux treated samples can be attributed to the dox-resistant cells or to the leftover POLE1. Since "dottiness" is a common artifact of DNA fiber and DNA combing experiments, quantitative analysis of purely "dotty" patterns can't be reliable. To test if the "dots" represent extremely slow replication forks, we tried labeling ongoing DNA synthesis in POLE1-depleted cells for 60 min (Supplementary Fig. 3d), however, the phenotype did not change and we did not observe any tracks that could indicate continuous DNA synthesis. One possible explanation of such "dotty" phenotype is excessive origin firing with replication failure shortly after.

In order to identify the proteins involved in DNA synthesis after POLE depletion, we performed an iPOND experiment[30] on clone 16 cells with and without dox/aux treatment for 16 h. As expected, the amount of nascent DNA and proteins in the samples from cells treated with dox/aux, was lower than in the control samples (Supplementary data 1). In order to compare the protein composition of the replisome with and without POLE1, we performed two alternative normalizations: (1) normalization to the signal of five major histones (intended to normalize to the amount of the DNA in the sample), (2) normalization to the signal of MCM subunits (intended to normalize to the number of replication forks in the sample). After normalization to the histones (Fig. 3f, Supplementary Fig. 3e), we observed a strong decrease of MCM subunits, subunits of all three replicative polymerases and all RPA subunits. After normalization to MCM subunits (Fig. 3g, Supplementary Fig. 3f, Supplementary Table 1), we still observed a strong decrease in both POLE1 and POLE2 subunits, implying that there existed a subset of active replication complexes that included MCM helicase, but lacked polymerase epsilon. POLE3 and POLE4 subunits' signals were very weak, and we therefore did not include them in the quantifications. We were also able to observe a decrease in POLA1 (polymerase alpha catalytic subunit), but DNA polymerase delta subunits showed no change relative to MCM after POLE1 depletion. This suggests that after POLE1 depletion there may exist a subset of active replication complexes that lack POLE, and possibly POLA. No other replication proteins showed a strong change in abundance relative to MCM after POLE1 depletion. No specialized polymerases were detected in the experiment. Overall, our data pointed to POLA/POLD-dependent DNA synthesis in the absence of POLE1.

## POLE1 depletion does not lead to MCM unloading from failed replication forks

A decrease in MCM signal compared to nascent DNA (histones) (Fig. 3f) could indicate CMG helicase unloading upon failure of replication in the absence of POLE—in this case nascent DNA would not be associated with an active helicase anymore. CUL2 and p97/VCP have been shown to control CMG disassembly during S-phase in mammalian cells[31], therefore we studied the signal for VCP/p97 in the iPOND data (CUL2 signal was insufficient for quantification). We observed an increase in VCP/p79 abundance in the iPOND pulldown, relative to MCM (Supplementary Fig. 4a), indicating possible active unloading of CMG through p97-mediated mechanism upon replication failure in the absence of POLE1. However, POLE1 depletion did not lead to a decrease of MCM7 on chromatin by FACS (Fig. 4a, b).

In order to study the presence of MCM at the sites of DNA replication more directly, we turned to single-molecule localization microscopy (SMLM) and analyzed EdU, PCNA, and MCM signals in replicating cells in high resolution (Fig. 4c). While EdU cluster density did not change after dox/aux treatment of clone 16 cells

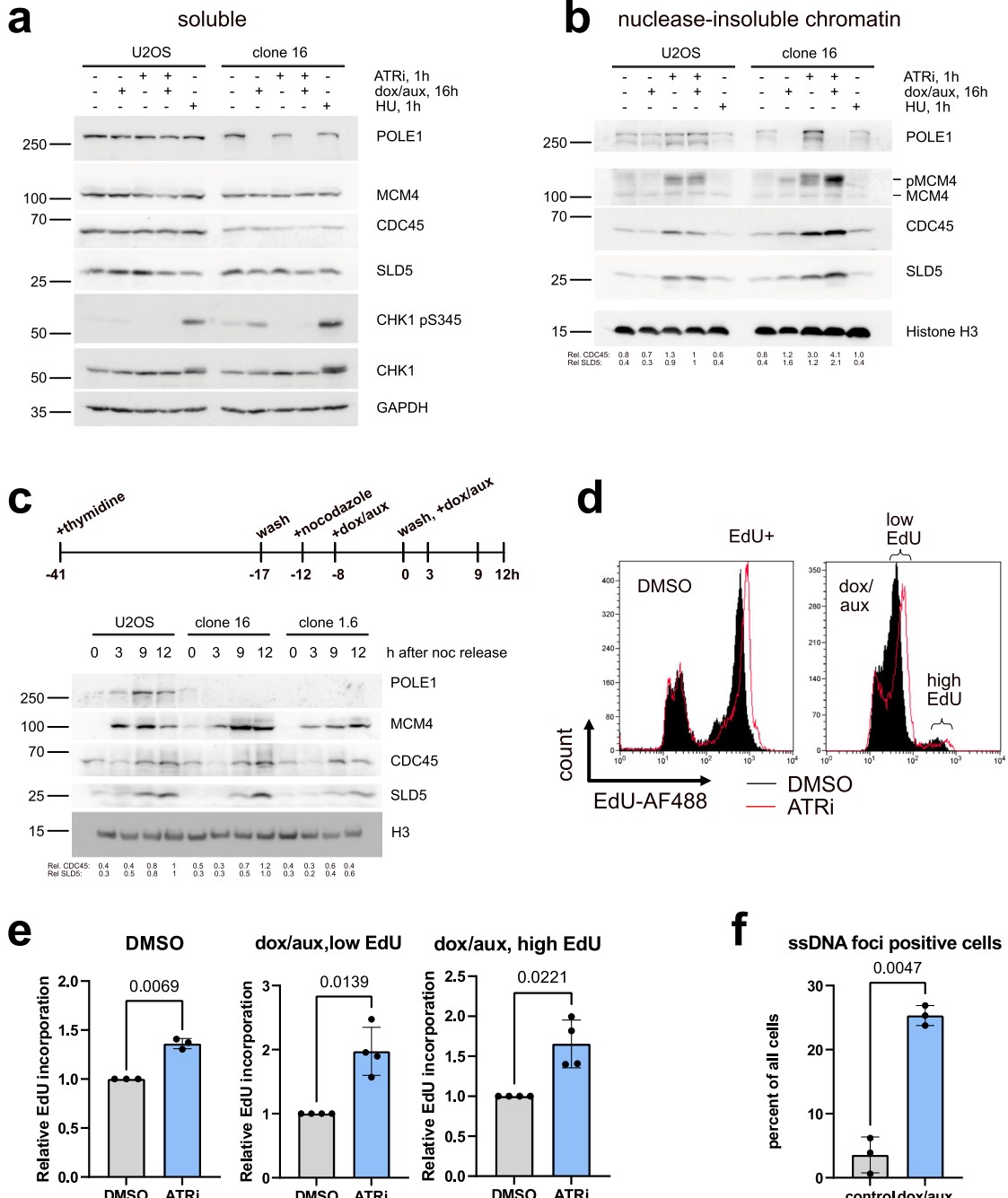

**Fig. 2 | Origin firing in POLE1-depleted cells. a**, **b** Wild-type U2OS or homozygous mAID-KI clone 16 were treated for 16 h with DMSO or dox/aux, 5 μM ATRi was added to the indicated samples for 1 h, followed by cell lysis and the isolation of the insoluble chromatin fraction. Equal amounts of protein were loaded. Western blot of the soluble lysates (**a**) and insoluble chromatin fraction (**b**) is shown. 1 h hydroxyurea (HU, 2 mM) treatment was used as a positive control for replication stress. The second band on POLE1 blot (**b**) likely represents partially degraded protein. Specific signals of SLD5 and CDC45 were quantified by Fiji/ImageJ. **c** Indicated cell lines were synchronized by thymidine/nocodazole blocks and treated with dox/aux as indicated. Western blot analysis of chromatin from the cells collected at the indicated timepoints is shown. Equal amounts of protein were loaded. Specific signals of SLD5 and CDC45 were quantified by Fiji/ImageJ.

**d**, **e** Clone 16 cells were treated for 16 h with DMSO or dox/aux as indicated, DMSO or 5 μM ATRi was added to the indicated samples for 60 min before harvest, 10 μM EdU was added for the last 30 min of treatment. Flow cytometry plots showing EdU incorporation histograms (**c**) and relative EdU incorporation, normalized to the samples without ATRi, are shown (**d**)−mean + SD from $n = 3$ (DMSO) or $n = 4$ (dox/aux) independent experiments. Paired t-test was used for statistical analyses, $p$ values are shown. **f** Clone 16 cells were incubated with 10 μM CldU for 48 h, DMSO or dox/aux were added for the last 16 h of treatment. After CSK extraction, cells were fixed and stained with anti-CldU antibodies under native conditions. Quantification of ssDNA-positive cells is shown−mean + SD from $n = 3$ independent experiments. Paired t-test was used for statistical analyses, $p$ value is shown. Source data are provided as a Source data file.

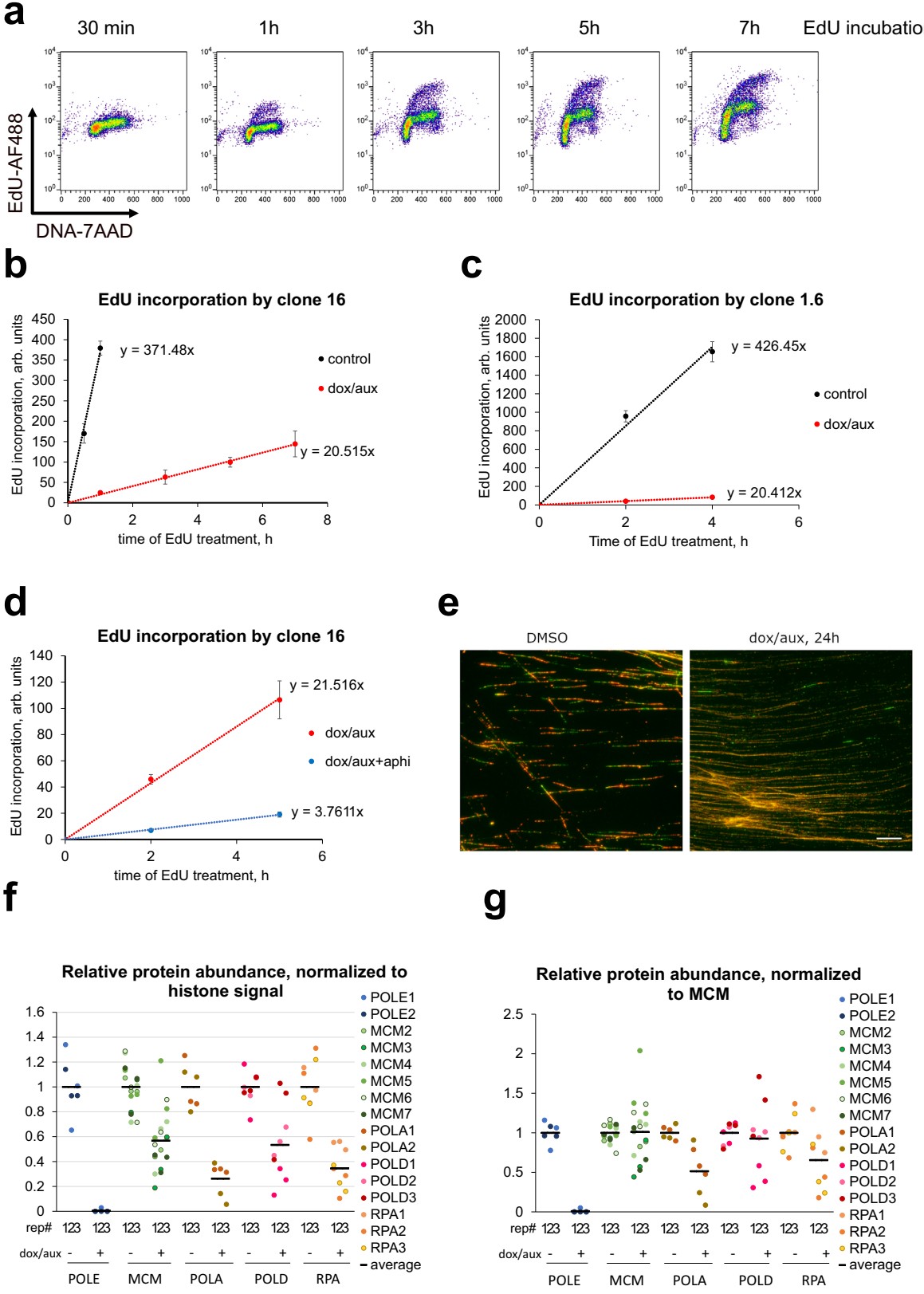

(Supplementary Fig. 4b), the amount of EdU per focus (Fig. 4d) and the average EdU density around PCNA (Supplementary Fig. 4c) significantly dropped after dox/aux treatment, confirming slow and inefficient DNA synthesis by POLE1-depleted cells. In contrast to EdU, both MCM cluster density (Supplementary Fig. 4d) and the amount of MCM per focus (Fig. 4e) were not affected by dox/aux treatment.

These data indicate that the failed origin firing in the absence of POLE1 does not lead to MCM unloading.

Interestingly, SMLM analysis of PCNA foci in POLE1-depleted cells showed that both PCNA cluster density (Fig. 4f) and the amount of PCNA per focus (Fig. 4g) slightly decreased in response to POLE1 depletion, while the overall PCNA signal remained

**Fig. 3 | The effect of POLE1 depletion on DNA synthesis. a–d** Clone 16 (**a**, **b**, **d**) or clone 1.6 (**c**) cells were treated for 16 h with dox/aux, followed by 10 μM EdU additional for the indicated times. For (**d**) 2 μM aphidicolin was added 1 h before the start of the EdU pulses. Flow cytometry plots showing EdU incorporation and DNA content (**a**) or EdU incorporation quantifications (**b**–**d**) are shown. Quantification is based on four (**b**) or three (**c**, **d**) independent experiments, means and standard deviations are shown, dox-resistant population was disregarded for the quantification. **e** Clone 16 cells were treated for 24 h with DMSO or dox/aux. Ongoing replication was labeled with 10 min pulse of CldU (red) followed by 20 min pulse of IdU (green) and visualized using DNA fiber analysis, as described in "Methods". Scale bar is 10 μm. **f**, **g** Clone 16 cells were treated for 16 h with DMSO or dox/aux, followed by 10 min EdU pulse and iPOND isolation of protein, associated with nascent DNA, and mass-spectrometry. The signal was normalized to average signal of histones (**f**) or MCM subunits (**g**) in each sample, and to respective DMSO-treated samples. The means from $n = 3$ experimental replicates for each group are shown as horizontal black lines. Source data are provided as a Source data file.

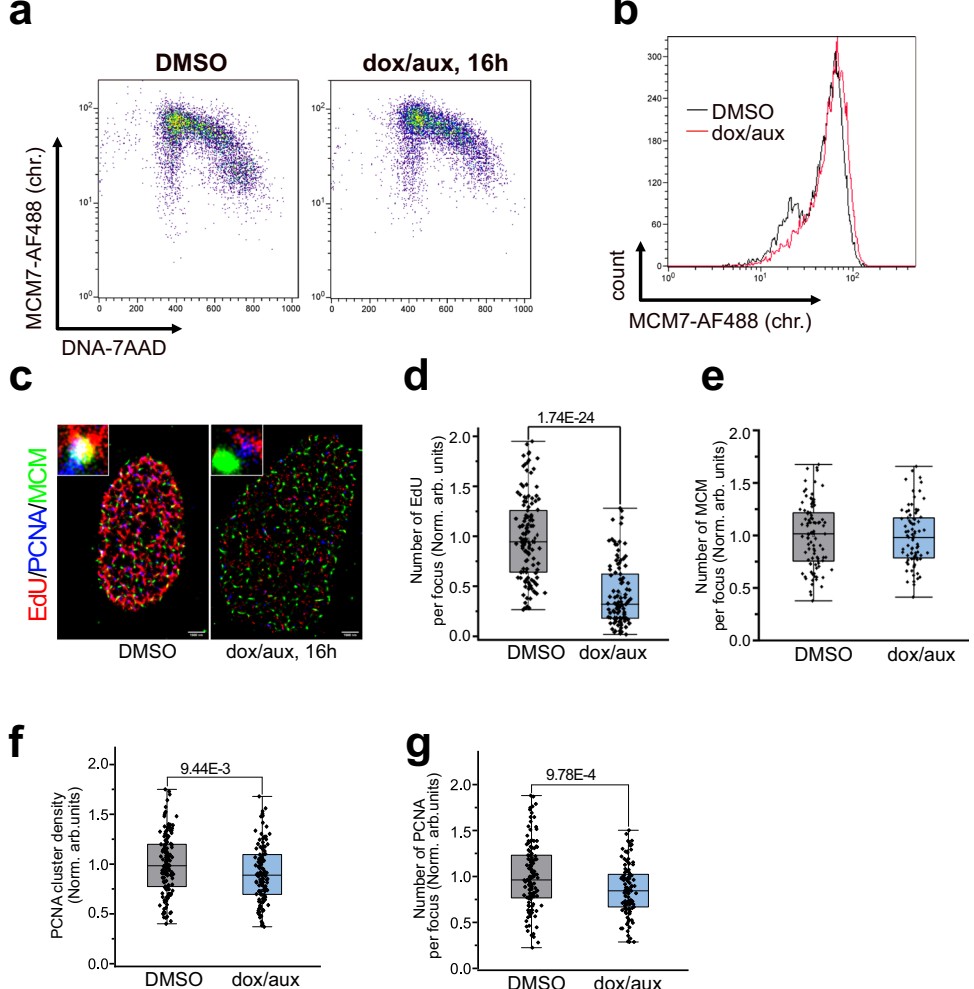

**Fig. 4 | MCM on chromatin after POLE1 depletion. a, b** Clone 16 cells were treated for 16 h with DMSO or dox/aux, followed chromatin extraction and MCM7 immunostaining. Flow cytometry plots of DNA/MCM7 staining (**a**) or MCM7 histograms (**b**) are shown. **c**–**g** Clone 16 cells treated for 16 h with DMSO or dox/aux were pulse labeled with thymidine analog EdU for 15 min prior to processing for super-resolution imaging. Representative multicolor super-resolution images (**c**) of PCNA (blue), MCM (green), and extent of EdU (red) incorporation in S-phase nuclei. Scale bar, 1500 nm. Quantitation of EdU (**d**) and MCM (**e**) detected per focus normalized to DMSO-treated clone 16 control, based on at least 3 independent experiments (For EdU per focus, $n = 131$, 109, and for MCM per focus $n = 95$, 83 for DMSO and dox/aux treated clone 16 cells. Student's t-test was used for analyses, mean + SD and the $p$ values are shown. Quantitation of PCNA detected per focus (**f**) and PCNA clusters (**g**), normalized to DMSO-treated clone 16 cells based on at least 2 independent experiments. (PCNA per focus, $n = 126$, 106; PCNA cluster density, $n = 139$, 120; student's t-test was used for analyses, mean + SD and $p$ values are shown). Source data are provided as a Source data file.

unchanged (Supplementary Fig. 4e). Lower number of PCNA molecules per fork is consistent with failure to establish lagging strand synthesis that normally harbors the majority of PCNA molecules.

### The C-terminal part of POLE1 supports DNA synthesis in the absence of the full-length protein
Previous studies in yeast have demonstrated that the non-catalytic C-terminal domain of POLE1 is sufficient for cell viability in the absence of the full-length protein, implying that the essential function of POLE in replication initiation is not its catalytic function[5,12]. In agreement with this, in *Xenopus laevis* egg extracts, the C-terminal non-catalytic domain of DNA polymerase epsilon was also able to partially restore DNA synthesis after the depletion of the endogenous protein[32]. In order to validate this model in human cells, we created several truncation mutants of POLE1 (Fig. 5a). Wild type and catalytically dead POLE1 were N-terminally tagged with FLAG-HA; N-terminal catalytic domain of POLE1 (cat), C-terminal domain (Δcat), and Δcat missing the

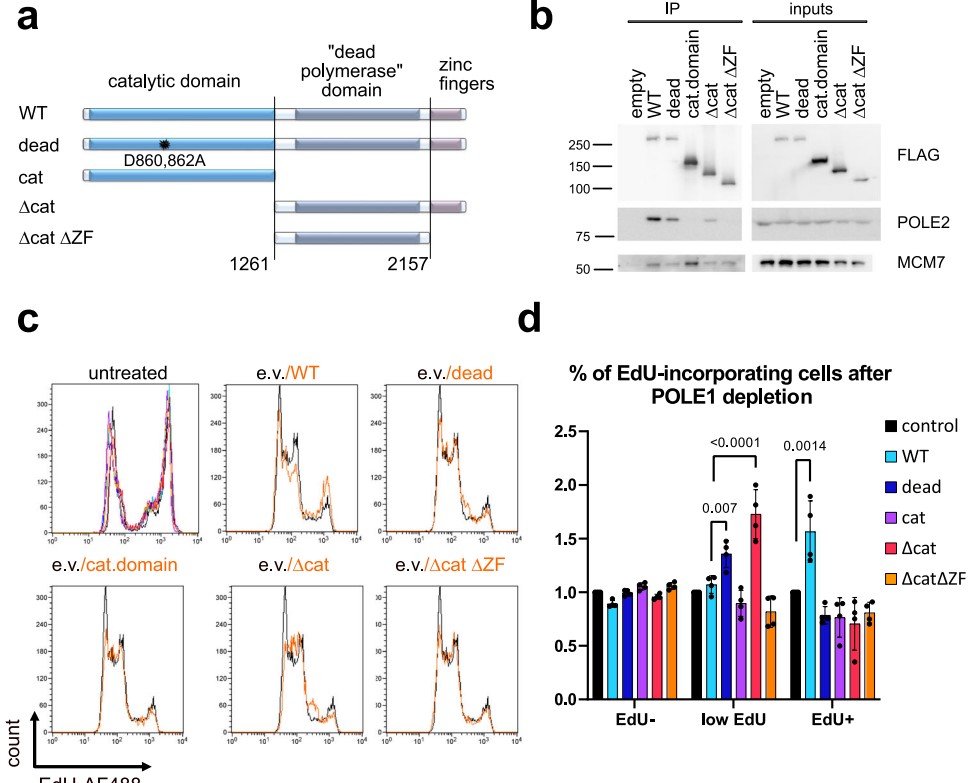

**Fig. 5 | C-terminal non-catalytic part of POLE1 is critical for replication initiation. a** Schematic representation of POLE1 constructs used in the study: wild-type protein, catalytically dead protein with mutations D860A, D862A, catalytic domain of POLE1 (aa1–1261), C-terminal half of POLE1 (without the catalytic domain, aa 1262–2305), C-terminal part of POLE1 without the zinc-finger domain (aa 1261–2151). **b** 293FT cells were transfected with an empty vector or the constructs described in panel a, fused to FLAG tags at their N termini. 48 h later cells were lysed and FLAG-tagged proteins were immunoprecipitated using M2 agarose beads, followed by elution with FLAG peptide. Western blots of the eluted protein and input samples are shown. **c, d** Clone 16 cells were transfected with indicated constructs. 32 h later dox/aux or DMSO was added to the cells for 16 h. 10 µM EdU was added for the last 30 min. Flow cytometry histograms of the EdU incorporation are shown (**c**). The first panel shows the transfected untreated samples, colors match the legend on 2D, the other panels represent samples treated with dox/aux for 16 h. Comparisons of samples transfected with indicated constructs (orange) to the sample transfected with an empty vector (black) are shown (**c**). **d** Quantification of the percent of cells in the dox/aux treated cells, corresponding to the indicated fractions, normalized to the empty vector control, are shown. Fractions were gated as indicated in Fig. S2. The quantification is based on *n* = 4 independent experimental repeats (means + SD and *p* values are shown where they are significant. One-way ANOVA was used for statistical analyses). Source data are provided as a Source Data file.

very C-terminal part containing zinc fingers (ΔcatΔZF) were N-terminally tagged with myc-FLAG tags.

In order to check if the mutations and deletions affected the interactions of POLE1 with other replication proteins, we expressed the constructs in 293FT cells, followed by immunoprecipitation with FLAG-M2 beads, elution with FLAG peptide, and western blotting with antibodies against POLE2 and MCM7 (Fig. 5b). Previous studies[33] have shown that the C-terminal zinc-finger region binds the second DNA polymerase epsilon subunit POLE2. In agreement with these data, we found that cat (1–1261) and ΔcatΔZF (1262–2157) constructs were unable to pull down POLE2. However, all the POLE1 constructs retained the ability to co-precipitate with MCM complex.

We then proceeded to test if any of the POLE1 constructs could rescue DNA synthesis after POLE1 depletion. 32 h after transfection of the clone 16 cells with the described POLE1 constructs, we treated cells with DMSO or doxycycline/auxin for 16 h followed by labeling of ongoing replication with EdU for 30 min, and FACS analysis. Our experiment showed that expressing POLE1 or its truncation mutants in the absence of dox/aux treatment did not affect the percentage of EdU-positive cells or the level of EdU incorporation (Fig. 5c). However, cells transfected with POLE1 mutants after the depletion of the endogenous POLE1, displayed differences in EdU incorporation profile (Fig. 5c, d). As expected, the expression of the WT construct led to the increase in the cells incorporating EdU at the level of untreated S-phase

cells, confirming that the WT protein can fully restore replication. The expression of the N-terminal catalytic domain of POLE1 had no effect on EdU incorporation by dox/aux treated cells, but the expression of the C-terminal domain of POLE or the catalytically inactive POLE1 led to an increase in the fraction of cells with low level EdU incorporation ("low EdU", gating is shown on Supplementary Fig. 5a). This fraction did not increase with the expression of the ΔcatΔZF fragment of POLE1. The quantification of the three fractions—EdU-negative cells, cells incorporating low levels of EdU, and cells incorporating normal (high) levels of EdU, normalized to the control transfected with an empty vector—is shown in Fig. 5d. These data indicate that, in the absence of the full-length protein, the C-terminal zinc-finger region of POLE1, responsible for the interaction with POLE2, is essential for the partial rescue of DNA synthesis by Δcat—the non-catalytic C-terminal domain of POLE1.

## Slow DNA synthesis in presence of POLE1 C-terminal non-catalytic domain

In order to further investigate the DNA synthesis supported by Δcat, we decided to develop a cell line that stably expressed Δcat and could degrade the endogenous POLE1 after dox/aux treatment. Unfortunately, clone 16 could not be used, due to its resistance to multiple antibiotics, which made additional selection for adding Δcat impossible. We therefore transfected clone 1.6 with a plasmid, expressing

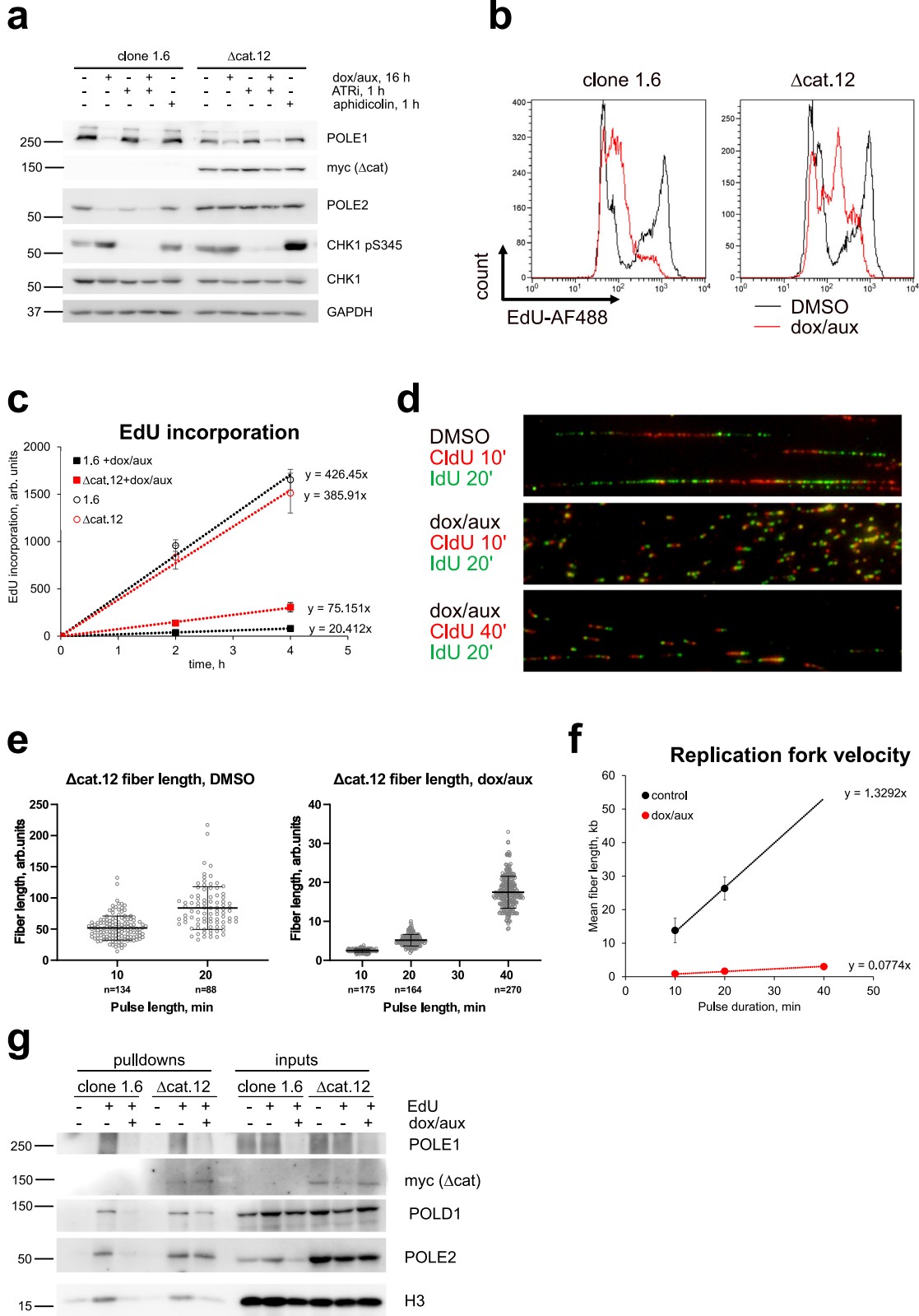

Δcat and a neomycin resistance marker, selected with G418, and performed single-cell cloning. We were able to identify two clones stably expressing Δcat–Δcat.12 and Δcat.25 (Fig. 6a, Supplementary Fig 6a). After the depletion of the endogenous POLE1 by dox/aux treatment, Δcat cells showed EdU incorporation about 5 times lower than that of the POLE1-proficient cells, but still 4 times higher than POLE1-depleted clone 1.6 cells (Fig. 6b, c, Supplementary Fig. 6b). As the selection/

subcloning took some time, the dox-resistant population in Δcat.12 and Δcat.25 is relatively high even in the very early passages (Fig. 6a, b, Supplementary Fig. 6a).

One of the key properties of Δcat is its ability to retain the interaction with POLE2 and MCM (Fig. 5b). Indeed, POLE1-depleted clone 1.6 showed a strong reduction in POLE2 level, indicating that POLE1 is necessary for POLE2 stability (Fig. 6a). Expressing Δcat completely

**Fig. 6 | DNA replication-dependent on the C-terminal non-catalytic domain of POLE1. a–c** Clone 1.6 homozygous mAID-KI cells (1.6) or clone 1.6 stably expressing myc-FLAG-Δcat - clone 12 (Δcat.12) were treated for 16 h with DMSO or dox/aux. **a** 5 μM ATRi was added to the indicated samples for 1 h. Western blot of the whole cell lysates is shown. 1 h aphidicolin (2 μM) treatment was used as a positive control for replication stress. **b**, **c** EdU was added for the last 30 min (**b**) or 2–4 h (**c**) of treatment, flow cytometry histograms of EdU incorporation (**b**) or EdU incorporation quantifications (**c**) are shown. Quantification is based on $n = 3$ independent experiments, means + SD are shown, dox-resistant population was disregarded for quantification. **d–f** Δcat.12 cells were treated for 16 h with DMSO or dox/aux. Ongoing replication was labeled with 10- or 40-min pulse of CldU followed by 20 min pulse of IdU and visualized using DNA fiber analysis, as described in "Methods". Representative images (**d**), individual fiber lengths from a representative experiment (mean and SD) (**e**), and mean fiber lengths (based on $n = 3$ experimental repeats) and SD of the means (**f**), are shown. **g** Clone 1.6 and Δcat.12 cells were treated for 16 h with DMSO or dox/aux, followed by 10 min EdU pulse (where indicated) and iPOND isolation of proteins, associated with nascent DNA. Western blot analyses of the inputs and pulldown samples are shown. Source data are provided as a Source data file.

rescued this effect, confirming that this C-terminal domain of POLE1 is sufficient for the interaction with and stabilization of POLE2 (Fig. 6a, Supplementary Fig. 6a).

Clones 16 and 1.6 exhibited ATR activation in response to POLE1 depletion (Figs. 2a, 6a, Supplementary Fig. 2b), however, Δcat expression rescued this phenotype, indicating that using Δcat in DNA replication did not lead to an accumulation of single-stranded DNA. ATRi treatment led to a modest increase in EdU incorporation in POLE1-depleted Δcat-expressing cells, suggesting lower availability of unfired dormant origins (Supplementary Fig. 6c, d).

In order to assess the replication dynamics of cells relying on Δcat in the absence of endogenous POLE1, we performed DNA fiber analysis (Fig. 6d–f, Supplementary Fig. 6e, f). Dox/aux treated Δcat cells showed a very distinct phenotype of extremely short DNA fibers (Fig. 6d). Using longer CldU pulses we were able to measure replication fork velocity in such cells, which was about 18 times lower than the mean fork speed in POLE1-proficient cells (Fig. 6f). While we were not able to reliably measure inter-origin distances in dox/aux treated cells, it is clear from the imaging (Fig. 6d) that the inter-origin distances in such cells are shorter. Based on 5× lower EdU incorporation and 18× lower replication fork velocity compared to POLE1-proficient cells, and given that EdU ∼ # of forks × fork speed, we can expect about 3.6× more replication forks in dox/aux treated 1.6 + Δcat cells. These data suggest that in U2OS cells the non-catalytic C-terminal domain of POLE1 is sufficient for DNA replication initiation, however, the resulting replication forks are extremely slow.

In the yeast system, in the absence of the catalytic domain of DNA polymerase epsilon, polymerase delta is thought to step in to synthesize the leading strand[14]. In order to check if this is true in human cells, we performed an iPOND experiment with clones 1.6 and Δcat.12 (Fig. 6f). Our data confirmed, that POLE2, Δcat, and POLD1 were present at the replication forks. These data are in agreement with the model where in the absence of POLE1 catalytic domain, its C-terminal non-catalytic domain was sufficient for the stabilization of POLE2 and DNA replication initiation, while POLD takes over the DNA synthesis at the leading strand.

In summary, our data indicate that in the absence of POLE1, replication origin firing in human cells proceeds past CMG assembly and activation, but fails at a later step. C-terminal non-catalytic domain of POLE1 is capable of rescuing this defect, resulting in continuous DNA synthesis, however, in this case replication forks are very slow.

## Discussion
### Origin firing in the absence of POLE1
In this study, we used an auxin-inducible degron system to establish rapid and efficient depletion of POLE1 in human cells. POLE1 knockdowns were previously used as controls in studies of other replication proteins[34,35]. In the study by Ercilla et al.[35] POLE knockdown only slightly decreased EdU incorporation by S-phase cells, indicating that knockdown efficiency may have not been sufficient. While POLE4 knockout led to a decrease in POLE1 concentration[21], the levels of POLE1 in the iPOND pulldowns were not affected by POLE4 knockout, confirming only partial depletion. In our study, the efficiency of rapid POLE1 depletion using an auxin-inducible degron system is confirmed

by its virtual absence in the iPOND pulldowns, which allowed us to observe and study defective replication origin firing in the absence of POLE1.

Here we show that while POLE1 depletion blocked any processive DNA synthesis, the number of active replication origins in POLE1-depleted cells was not limited by the level of POLE1, as additional origins were rapidly activated by ATR inhibition, which can be observed by the recruitment of the replication proteins to the chromatin fraction, phosphorylation of MCM4, and the increase in EdU incorporation (Fig. 2a, b, d). CMG assembly on chromatin was also observed during S-phase entry of POLE1-depleted cells (Fig. 2c).

One possible explanation of these data is that the scarce POLE transiently associates with pre-RCs, ensuring replication initiation, and quickly dissociates, moving on to the next pre-RC. This would agree with the absence of POLE1 in the iPOND pulldowns, but it would contradict a recent study[36] showing that yeast DNA polymerase epsilon has a very low exchange rate at the replication fork in vitro and this rate only goes down with the decrease of the concentration of polymerase epsilon. An alternative explanation is that POLE1 is not required for the initial steps of DNA replication initiation, but DNA synthesis stalls quickly after origin firing.

In yeast estimated the initial DNA synthesis at the origins (-180 bp) is performed by DNA polymerase delta[2], after which polymerase epsilon takes over. We propose that the DNA synthesis that we observe in the absence of POLE1 is the initial POLA/POLD-dependent synthesis that fails at the polymerase switch step (Fig. 7)—one notable structural rearrangement after the initial helicase activation. While further biochemical and structural studies are needed to confirm this model, it is supported by a "dotty" fiber pattern (Fig. 3e) and aphidicolin sensitivity of EdU incorporation by POLE1-depleted cells (Fig. 3d). Based on yeast data, we expected 100–200 bp tracks synthesized by POLD before the switch[2,4], however, at 2.59 kb/μm[37] 100 bp tracks would appear as 39 nm. This is below the resolution limit of our microscope, so we believe that such short DNA pieces appear as "dots" in the DNA fiber analysis.

Our data indicate that after DNA replication fails, MCM helicases are not unloaded and remain associated with chromatin (Fig. 4a–e). In addition, based on the accumulation of single-stranded DNA and ATR activation in POLE1-depleted cells (Fig. 2a, f), it is likely that MCM continues DNA unwinding for some time after DNA synthesis fails. It remains unclear what happens to these failed replication complexes, but being located far enough from the failed DNA synthesis could explain their decrease in the iPOND pulldown. As a standard response to ssDNA and replication stress, we would expect the recruitment of fork stabilization and remodeling proteins[38], but since proper replication forks are never established in the absence of POLE, the same mechanisms may not apply. No pathway allowing activated MCMs to be re-loaded back onto double-stranded DNA has been described to date, so MCMs from failed replication initiation likely remain encircling one DNA strand each.

A recent series of publications[31,39,40] demonstrated that the DNA strand excluded by MCM helicase masks the site on MCM required for its recognition by CUL2/LRR ubiquitin ligase, preventing the disassembly of CMG helicase. In our model, where replication can't be

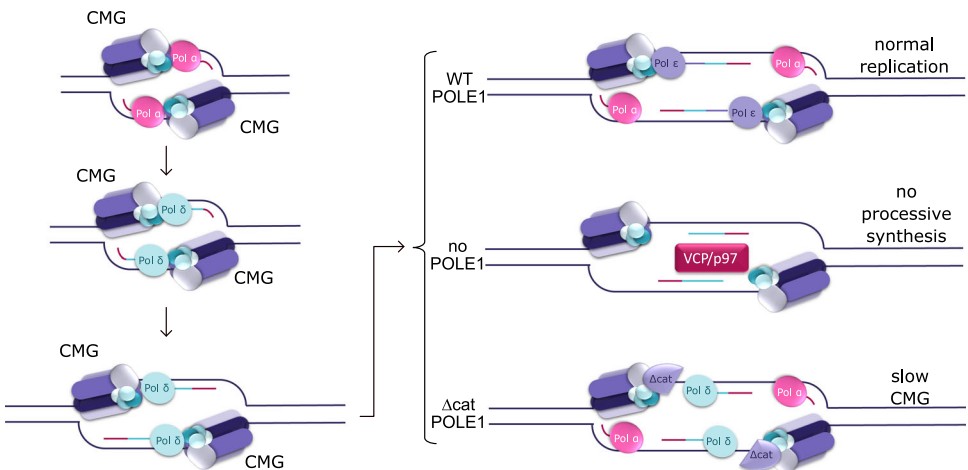

**Fig. 7 | Proposed model of DNA replication initiation in human cells with and without POLE1, as well as in presence of Δcat.** POLE may be present at the early stages of origin firing, but according to our data it is not necessary. Under normal conditions, POLE replaces POLD as a leading strand polymerase. In the absence of POLE, DNA synthesis fails at the polymerase switch step. Non-catalytic C-terminal domain of POLE1 is sufficient to proceed to replication elongation, in this case, POLD assumes the role of the leading strand polymerase, but the speed of DNA synthesis is much slower.

completed and MCM encircles one strand of DNA, allowing the excluded strand to prevent its disassembly, any unloading of the activated CMG from DNA is highly unlikely. A slight increase in p97 association with nascent DNA (Supplementary Fig. 4a) could be related to its recently proposed role in unloading excessive POLA/PRIM during origin firing[41], but this requires further investigation.

### DNA synthesis in the presence of C-terminal non-catalytic domain of POLE1

In U2OS cells that lack full-length POLE1, the expression of the C-terminal non-catalytic domain of POLE1 was sufficient for continuous albeit slow DNA synthesis (Fig. 6d, e). Similar observations have been previously made in yeast[13], and our data suggest that the non-catalytic function of DNA polymerase epsilon is conserved in mammalian cells. Furthermore, in yeast, in the absence of the catalytic domain of Pol2, polymerase delta replicated both strands[14]. We observe a similar phenotype in U2OS cells expressing Δcat in the absence of the full-length POLE1: POLD is associated with nascent DNA together with POLE2 and the C-terminal domain of POLE1 (Fig. 6f).

In the absence of POLE1 catalytic domain in U2OS cells, replication fork velocity was dramatically reduced (Fig. 6d, e). Nevertheless, such cells did not show any strong ATR activation, implying the absence of any significant replication stress caused by helicase/polymerase uncoupling. Therefore, the slow replication fork velocity is likely caused by slow DNA unwinding by the CMG helicase. Similar observations were previously made in a reconstituted yeast system[17], confirming that the important role of the N-terminal catalytic domain of DNA polymerase epsilon catalytic subunit in full activation of the CMG helicase is conserved.

### Protein-protein interactions of POLE1 and their role in origin firing in human cells

Our data demonstrate that the interaction between POLE1 and POLE2 was essential for supporting DNA synthesis: deleting the C-terminal zinc-finger region of Δcat, responsible for the interaction with POLE2[33] (Fig. 5b), completely abolished its ability to support EdU incorporation in the absence of the full-length protein (Fig. 5c, d). Moreover, the expression of Δcat was necessary and sufficient for POLE2 stability and recruitment to the replication fork (Fig. 6f). These data indicate that the essential role of POLE1 in replication initiation may include the stabilization and recruitment of POLE2. What makes POLE2 presence at

the replication initiation sites critical, remains to be elucidated. In yeast, the second subunit of DNA polymerase epsilon Dpb2 is essential for CMG assembly, and the expression of its N-terminal domain was sufficient to support cell viability, producing replisomes that lack DNA polymerase epsilon[7]. However, according to our data, POLE1 and POLE2 (which is destabilized in the absence of POLE1) are dispensable for CMG assembly and MCM phosphorylation on chromatin in response to ATRi in human cells (Fig. 2b, c). We therefore propose that POLE2 is critical at one of the later steps of replication initiation, possibly at the structural perturbations associated with the polymerase switch. According to a recent structural study[42], human POLE2 binds the GINS-MCM junction of the CMG helicase, which could make POLE2 essential for the CMG stability during some conformational changes, such as the polymerase switch step. Further biochemical and structural studied are necessary to address this question.

We found that both N-terminal catalytic domain and C-terminal non-catalytic domain of POLE1 can co-precipitate with MCM (Fig. 5b). According to structural studies of yeast[43] and human[42] replisomes, the PolE active conformation is connected to CMG solely through the C-terminal part of Pol2. However, the N-terminal domain of Pol2 participated in CMG binding in the inactive conformation[5]. In agreement with these findings, the N-terminal domain of Pol2 has recently been shown to play a role in mediating CMG-PolE interaction in yeast[44]. Our data suggest a possibility for a similar mechanism in human cells.

Overall, our study provides some insights into the non-catalytic role of DNA polymerase epsilon in human cells. Since POLE1 mutations associated with multiple cancers and developmental diseases are often outside its catalytic domain, elucidating the non-catalytic functions of this protein may help shed a light on the molecular mechanisms behind these diseases.

## Methods
### Plasmids and cloning
For osTIR1 KI we used PX458-AAVS1 (a gift from Adam Karpf—Addgene plasmid # 113194) and pMK243-Tet-OsTIR1-PURO (a gift from Masato Kanemaki—Addgene plasmid # 72835) plasmids[24]. For mAID KI templates, homology arms were synthesized by Genscript in the pUC57 vector with stop codon substituted to BamHI site (TGCCTCGGC TCAGCCTGGCCTCCTTGGCCTCCTCTCTGAGTGGACTGGGGTCTCAC TGTGCCTGTTTATTCCTGCAGGTCTTCATGGAACAGATCGGAATATTC CGGAACATTGCCCAGCACTACGGCATGTCGTACCTCCTGGAGACCCT AGAGTGGCTGCTGCAGAAGAACCCACAGCTGGGCCATGGATCCCCAG

CCCCGGGCCCCGGGTGCCTCTGCGTCCGTGCCAGGCCTCCTGATGCC
AAGGCCACATCCCCGTGCTTCCAGTGACCAGACCACTGACCACCCTG
ACTGTCCAAACCTGTGACCCCAGGCCAGGGAACGGGGAGGAAACCA
AAG).

Inserts from plasmids pMK293 (mAID-mCherry2-Hygro) and pMK292 (mAID-mCherry2-NeoR) (Addgene plasmids # 72831 and # 72830 were gifts from Masato Kanemaki)[24] were cloned into the BamHI site of the synthesized plasmids. POLE1 C-terminus-targeting gRNA (GTCGUACCUCCUGGAGACCC) was expressed from pSpCas9 BB-2A-Puro (PX459) v2.0 plasmids (Genscript). POLE1 plasmid[45] was used as a template for creating the described deletion mutants, which were cloned into pCMV-AN-myc-DDK vector (Origene). Catalytically dead POLE1 was described previously[45].

## Cell lines, cell culture, and transfections
U2OS (ATCC HTB-96) cells were grown in RPMI media (Lonza), supplemented with 10% FBS (GIBCO) and 1% penicillin-streptomycin (Invitrogen). For KI cells were transfected with corresponding gRNA and one or two (NeoR and HygroR for mAID-mCherry KI) HR templates. Growth medium was changed 8 h after transfection, 2.5 μM DNAPK inhibitor was added for 48 h. KI cells were selected with G418, puromycin and/or hygromycin until the non-transfected control died, followed by single-cell cloning and KI validation by PCR and/or western blot. Transfections were carried out using Lipofectamine 2000 (Thermofisher), according to manufacturer's instructions.

293FT cells (Thermofisher #R70007) were grown in DMEM media (Lonza), supplemented with 10% FBS (GIBCO) and 1% penicillin-streptomycin (Invitrogen).

## Cell lysis, insoluble chromatin isolation, and western blots
Cells were lysed in 50 mM Tris·HCl (pH 7.5), 150 mM NaCl, 50 mM NaF, 1% Tween-20, 0.5% Nonidet P-40, and protease inhibitors (Pierce #A32953) for 20 min on ice. Lysates were cleared by centrifugation, and soluble protein was used for immunoprecipitation or mixed with 2× Laemmli Sample Buffer (Bio-Rad) and incubated for 7 min at 96 °C and analyzed by Western blot. For nuclease insoluble chromatin, pellets were suspended in 150 mM Hepes (pH 7.9), 1.5 mM MgCl$_2$, 10% glycerol, 150 mM potassium acetate, and protease inhibitors containing universal nuclease for cell lysis (ThermoFisher, #88700) and incubated for 10 min at 37 °C on the shaker. Nuclease-insoluble chromatin was pelleted by centrifugation, washed with water, and dissolved in Laemmli Sample Buffer.

For western blot analyses, proteins were separated in 8–12% SDS-polyacrylamide gels in Running buffer (25 mM Tris, 192 mM glycine, 0.1% SDS), transferred onto PVDF membrane (BioRad, #1620177) in transfer buffer (25 mM Tris, 192 mM glycine,10% ethanol), blocked with 5% non-fat milk (BioRad, #1706404) in TBST (Thermofisher, #BP2471, 0.1% Tween-20), incubated with an appropriate dilution of the primary antibody overnight at 4 °C, washed with TBST buffer, incubated with secondary antibody for 1 h at room temperature, washed with TBST and developed using SignalFire™ Elite ECL Reagent (Cell Signaling, #12757P) and ImageQuant LAS 4000 imager (GE Healthcare). Quantification of western blots was performed using Fiji/ImageJ (version 1.53u).

## Synchronizations and chromatin loading of CMG components
2 mM thymidine was added to ~25% confluent cells for 24 h. After thymidine removal, cells were washed once with warm PBS and allowed to recover in fresh medium for 5 h. Nocodazole was then added for 12 h to stop the cells in G2/M. Dox/aux were added 4 h after nocodazole to ensure the completion of DNA replication before POLE1 depletion. 8 h later, cells were released from nocodazole, washed once with warm PBS and incubated in pre-warmed medium with dox/aux for the indicated periods of time.

In order to assess the loading of the CMG components on chromatin, samples were collected by trypsinization at the indicated timepoints, pellets washed once with ice-cold PBS and kept at −80 °C, and processed the next day.

Thawed pellets were resuspended in CSK buffer (10 mM PIPES pH 7.0, 300 mM Sucrose, 100 mM NaCl, 3 mM MgCl$_2$, 0.5% Triton X-100, protease inhibitors (Pierce #A32953)), incubated for 5 min on ice, followed by a 5-min centrifugation (1000 × g, 4 °C). The supernatant was collected as "soluble fraction", the pellets were washed once more with CSK buffer, digested in CSK buffer with universal nuclease for cell lysis (ThermoFisher, #88700) for 10 min at 37 °C. Samples were mixed with 2x Laemmli Sample Buffer (BioRad) and boiled for 10 min before proceeding to western blot analysis.

## Antibodies
POLE1 (Santa Cruz, #sc-390785, 1:500), osTIR1 (MBL International, #PD048, 1:1000), GAPDH (Santa Cruz, #sc-47724, 1:1000), pCHK1 (Cell Signaling, #2360S, 1:1000), Chk1 (Cell Signaling, #2348S, 1:1000), MCM4 (Cell Signaling, #3228S, 1:300), CDC45 (Santa Cruz, #sc-55569, 1:500), SLD5 (Santa Cruz, #sc-398784, 1:300), H3 (Santa Cruz, #sc-517576, 1:1000), MCM7 (Santa Cruz, #sc-9966, 1:1000), POLE2 (Santa Cruz, #sc-398582, 1:500), PCNA (Santa Cruz, #sc-56, 1:1000), FLAG (Sigma, F3165-1MG, 1:3000), myc (Cell Signaling, #2276, 1:1000).

## Flow cytometry
For EdU FACS, cells were treated with 10 μM EdU for 10 min, trypsinized, washed with PBS, and fixed with cold 70% ethanol on ice for 30 min to overnight. Cells were washed with PBS, and EdU staining was performed by using the EdU Click-iT kit (Thermofisher, # C10632), according to the manufacturer's instructions. For DNA staining, we used 7-AAD (7-Aminoactinomycin D) (Thermofisher, # A1310) or FxCycle™ PI/RNase Staining Solution (Thermofisher, #F10797).

Chromatin association of MCM was assessed essentially as described[46]: after trypsinization and PBS wash, cells were extracted with CSK buffer (10 mM PIPES pH 7.0, 300 mM Sucrose, 100 mM NaCl, 3 mM MgCl$_2$, 0.5% Triton X-100, protease inhibitors (Pierce #A32953), fixed with 4% paraformaldehyde, and blocked with 5% BSA followed by immunostaining with anti-MCM7 antibody (Santa Cruz, #sc-9966) at 1:200 dilution. 7-AAD (7-Aminoactinomycin D) (Thermofisher, # A1310) was used for DNA staining.

Flow cytometry was performed on an FACSCalibur flow cytometer, and data were analyzed by using FCSalyzer software. GraphPad Prism 9 was used for statistical analyses.

## Fiber analysis
DNA fiber analysis was performed essentially as in ref. 47. Briefly, ongoing DNA synthesis was labeled with the indicated nucleotide analogs, cells were washed with PBS, lysed with lysis buffer (0.5% SDS, 200 mM Tris−HCL (pH 7.4) and 50 mM EDTA) and spread on glass slides by tilting. After drying, the slides were fixed in methanol: acetic acid (3:1), dried, and rehydrated in PBS. DNA was denatured by incubating the slide in 2.5 N hydrochloric acid for 1 h. After neutralization in PBS, samples were blocked in blocking buffer (10% NGS in PBS), and stained with primary antibodies (Mouse Anti-BrdU Clone B44 (BD, #347580) for IdU, Abcam ab6326 for CldU, both 1:50 in the blocking buffer), washed with PBS-1%Tween20, incubated with fluorescently labeled secondary antibodies (Goat anti-mouse AlexaFluor 488 (Invitrogen, #A-11001), Goat anti-rat AlexaFluor 594 (Invitrogen, #A-11007), both 1:150 in blocking buffer). After extensive washes with PBS-Tween20 and PBS, slides were mounted with Prolong Diamond Antifade mounting medium (Invitrogen, #P36961). Samples were imaged using Olympus BX61 fluorescence microscope at 60x magnification; image analysis was performed using Fiji (ImageJ) software. At least 100 tracks per sample were analyzed.

## PCR

For validations of the knock-ins, genomic DNA was isolated using genomic DNA miniprep kit (Zymo Research, # D3025). Primers: Endogenous allele (C-terminus of POLE coding region) Forward GACCAGCATGCCTGTGTACTG, Reverse CTCCCTCCTGTGACGTCTGAG; mAID KI Forward GACCAGCATGCCTGTGTACTG, Reverse GCGGCATGGACGAGCTGTACAA; osTIR1 endogenous allele (AAVS) Forward GGTCCGAGAGCTCAGCTAGT Reverse TGGCTCCATCGTAAGCAAAC; primers to amplify tetR promoter (used for cloning) Forward: GGTACCGAGGAGATCTGCCGCCGCGATCGCGCGCCCTGGTTTACATAAGCAAAGCTTATA, Reverse: ATCGTCGTCATCCTTGTAATCCAGGATATCGTCCAGTCTAGACATGGTAATTCGATGATC.

## iPOND

Nascent DNA pulldown was performed essentially as described[30]. Four 150 mm dishes were used for each sample. Three independent experiments were performed. Briefly, ongoing DNA synthesis was labeled by incubation with 10 μM EdU for 10 min, followed by washes and fixation with 1% formaldehyde, quenched with 0.125 M glycine, scraped off the plates and permeabilized with 0.25% Triton X-100 in PBS. After washes with PBS-0.5% BSA, click reaction was used to label EdU with biotin-azide (25 μM azide-PEG3-biotin (Sigma # 762024), 10 mM sodium ascorbate, 2 mM copper sulfate). After washes with PBS-BSA cells were lysed in RIPA buffer (150 mM NaCl, 50 mM Tris−HCl pH 7.5, 1% Triton X-100, 0.1% SDS, protease inhibitor cocktail (Pierce #A32953) DNA was sheared using Bioruptor (Diagenode), 20 cycles of 30 s on/30 s off. After 15 min centrifugation (14,000 × g, 15 min), supernatants were incubated with streptavidin agarose (Sigma Merck #S1638) overnight. After 3 washes with RIPA buffer, samples on agarose beads were stored at −80 °C and transported to the Tartu University Proteomics Facility on dry ice. Alternatively, the beads were boiled with Laemmli Sample Buffer for 30 min, followed by western-blot analysis.

Proteomics sample preparation and LC/MS/MS analysis was performed at Proteomics core facility at the University of Tartu, Estonia, essentially as described previously[48], except that identification was carried out against UniProt Homo sapiens reference proteome with POLE1-mAID-mCherry addition. The detailed description is available alongside the deposited proteomics dataset with the ProteomeXchange ID PXD033757.

To assess the abundance of replication proteins we normalized the intensity of each protein of interest to the sum intensity of 6 MCM subunits or sum intensity of major histones (H1, H2A, H2B, H3, H4) in each sample, as indicated. Normalization approach was based on previously published method[49]. In order to disregard the differences in intensities between proteins (as they largely depend on the protein's physical properties), we normalized the data for each protein to the average intensity of this protein in the control samples, focusing on the changes induced by treatments (except Supplementary Fig. 4a showing data for one protein only).

## Single-stranded DNA staining

In order to assess the presence of ssDNA in the cells, cells were treated with 20 μM CldU for 48 h, dox/aux were added to the indicated samples for the last 16 h of treatment. After two PBS washes, cells were briefly extracted with CSK buffer (10 mM PIPES pH 7.0, 300 mM Sucrose, 100 mM NaCl, 3 mM MgCl$_2$, 0.5% Triton X-100; 5 min on ice), followed by fixation with 4% paraformaldehyde for 10 min at room temperature. After washes with PBS, samples were blocked with 5% BSA and stained with anti-BrdU antibody (ab6326, 1:150) followed by the incubation with secondary antibody, goat anti-rat AlexaFluor 594 (Invitrogen #A-11007, 1:300). After extensive PBS washes, nuclei were stained with 0.1 μg/ml Hoechst 33342 for 10 min and after a brief wash with PBS, samples were mounted using Prolong Diamond Antifade mounting medium (Invitrogen, #P36961). Samples were imaged using Olympus BX61 fluorescence microscope at 60x magnification; cell containing over 5 bright ssDNA foci were considered ssDNA-positive.

## Sample preparation, imaging, and image analysis for SMLM

Clone 16 cells were seeded onto glass coverslips (Fisher Scientific, 12-548-B) in a 6-well plate. At 50% cell confluence, cells were incubated with 2 μg/ml doxycycline (dox) and 500 μM auxin (aux) for 16 h to promote POLE1 degradation. Cells were incubated with 10 μM EdU for 15 min prior to harvest to detect nascent DNA synthesis. Cells were then permeabilised with 0.5% Triton X-100 in ice-cold CSK buffer (10 mM Hepes, 300 mM Sucrose, 100 mM NaCl, 3 mM MgCl$_2$, pH 7.4) for 10 min at room temperature to remove a majority of the cytoplasmic and non-chromatin bound proteins followed by three PBS washes. Cells were fixed with 4 % paraformaldehyde (Electron Microscopy Sciences, 15714) in PBS for 15 min, at room temperature. After washing twice with PBS, cells were washed with blocking buffer for 5 min, thrice (2% glycine, 2% BSA, 0.2% gelatine, and 50 mM NH$_4$Cl in PBS). Incorporated EdU was detected using the Click-iT Plus EdU Alexa Fluor 647 Imaging Kit (ThermoFisher, C10640). Cells were then washed with blocking buffer for 5 min, thrice followed by overnight incubation in blocking buffer at 4 °C. Cells were then incubated with primary antibodies against MCM6, (Abcam ab236151, conjugated to IR750 – cat #1556-1, 1:2500) and PCNA (Santa Cruz Biotechnology, sc-56, 1:1000) in blocking buffer for 1 h, at room temperature. After three washes for 5 min each with blocking buffer, cells were incubated with mouse IgG, AF 488 (Invitrogen, A11029, 1:5000) in blocking buffer for 30 min, at room temperature. After three 5 min washes with blocking buffer, stained coverslips were mounted onto a glass microscope slide and freshly prepared imaging buffer (1 mg/mL glucose oxidase (Sigma, G2133), 0.02 mg/mL catalase (Sigma, C3155), 10% glucose (Sigma, G8270), 100 mM mercaptoethylamine (Fisher Scientific, BP2664100)) was flowed through just prior to imaging.

For SMLM-SR imaging, raw image stacks with a minimum of 2000 frames for each color, acquired at 33 Hz, were captured on a custom-built optical imaging platform based on a Leica DMI 300 inverted microscope with three laser lines, 488 nm (Coherent, Sapphire 488 LPX), 639 nm (Ultralaser, MRL-FN-639-1.2), and 750 nm (UltraLaser, MDL-III-750-500). Lasers were aligned and combined using dichroic mirrors and focused onto the back aperture of an oil immersion objective (Olympus, UApo N, 100x, NA = 1.49, TIRF) via multiband dichroic mirror (Semrock, 408/504/581/667/762-Di01). For multicolor imaging, fluorophores were sequentially excited using a Highly Inclined and Laminated Optical (HILO) illumination configuration. Corresponding emissions were expanded with a 2X lens tube, filtered using single-band pass filters in a filter wheel (ThorLabs, FW102C): IR750 (Semrock, FF02-809/81), AF488 (Semrock, FF01-531/40), AF647 (Semrock, FF01-676/37) and collected on a sCMOS camera (Photometrics, Prime 95B). A 405 nm laser line (MDL-III-405-150, CNI) was used to drive Alexa Fluor 647 fluorophores back to their ground state. Images were acquired using Micro-Manager (v2.0) software.

Precise localization of each collected single molecule was performed as described in refs. 50−53. For display purpose, the representative images were generated by rendering the raw coordinates into 10 nm pixel canvas, convolved with a 2D-Gaussian ($\sigma = 10$ nm) kernel and brightness/contrast of the individual color channels were adjusted for display purposes. Raw result tables of the localization coordinates for each fluorophore blinking within a $6 \times 6$ μm$^2$ square region of interest (ROI) were directly submitted to Auto-Pair-Correlation analyses[54,55] to estimate the nuclear density of the fluorophores. Any artificial blinking events (one blinking event recorded multiple times in consecutive frames) were eliminated before the computation of cross-pair correlations.

A correlation profile was plotted as a function of the pair-wise distances and fit into a Gaussian model. The average molecular content and density within each focus was derived based on the computed

average probability of finding a given species around itself and the apparent average radius of the focus. This function estimated the nuclear density of EdU and MCM fluorophores within a nucleus, as well as average number of fluorophores within each EdU or MCM focus. For the Cross-PC analyses, the correlation profile was plotted as a function of the pair-wise distances between EdU and PCNA, and fitted into a Gaussian model to determine the cross EdU-PCNA pair correlation amplitude. Using this, we were able to estimate the average local density of EdU around each PCNA molecule within a given ROI.

### Reporting summary

Further information on research design is available in the Nature Portfolio Reporting Summary linked to this article.

## Data availability

The mass spectrometry proteomics data generated in this study have been deposited to the ProteomeXchange Consortium (http://proteomecentral.proteomexchange.org) via the iProX partner repository[56] with the dataset identifier PXD033757. The UniProt Homo sapiens reference proteome data used in this study for mass-spectrometry data analysis are available in the Uniprot database under Proteome ID UP000005640. Source data are provided with this paper.

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

## Acknowledgements

This work was supported by Estonian Research Council (research grant PRG1477 to T.N.M). Research in E.R. lab is supported by NIH grants: R35 GM134947, AI153040, and CA247773 (E.R.), the V Foundation BRCA Research collaborative grant (E.R.) and by Pfizer (E.R.). We are grateful to Dr. Julieta Martino, Dr. Peter Ly, and Alison Guyer for advice on fiber spreading assay. We thank Dr. Sergo Kasvandik for the advice on the analysis of mass-spectrometry iPOND data.

## Author contributions

T.N.M. conceived the project. T.N.M., D.G., S.J., and E.R. designed the experiments. S.V., H.A., and T.N.M. performed and analyzed all experiments except SMLM which were performed and analyzed by D.G. and S.J. T.N.M. wrote the manuscript with review and editing from S.V., D.G., S.J., H.A., and E.R.

## Competing interests

The authors declare no competing interests.
