## [Peer Review File · Nature Communications]

The non-catalytic role of DNA polymerase epsilon in replication initiation in human cellsREVIEWER COMMENTS

Reviewer #1 (Remarks to the Author):

DNA polymerase epsilon is critical replication polymerase that is responsible for normal leading strand synthesis. Older studies from yeast showed that Pol epsilon was dispensable for normal cell growth while Pol delta, the primary lagging strand pol was essential. Newer results have placed Pol epsilon interacting with the CMG helicase and actually taking over leading strand synthesis after initiation via Pol alpha/Pol delta. Whether this model holds in mammalian cells has not been directly tested. The current study begins to address these issues by testing a number of replication and DNA synthesis outcomes using an auxin-dependent degron approach. They make several notable and interesting findings. Removing 90% of pol epsilon catalytic subunits dramatically impairs cell proliferation and DNA synthesis. However there are large numbers of focus-like DNA synthesis events by DNA fiber analyses. Adding the Zinc finger-containing C-terminal fragment can partially restore shorter DNA synthesis events, suggesting that the Pol epsilon Cterm-CMG interaction may stabilize the replication initiation enough to start more DNA synthesis. Overall this study will be of significance to the field since it confirms and extends the model of initial DNA synthesis proposed in yeast and demonstrates the utility of a new tool for polymerase protein reduction.

Major Comments

Based on the 72 hr growth curve and described passaging issues the C-terminal AID-mCherry is likely impairing POLE function even in the absence of auxin-induced degradation. It is also unclear what the role of the C-terminal mCherry is. If the deltaCatalytic-mCherry construct suppressed the EdU incorporation defect to the same extent as the deltaCatalytic construct, this would clarify what is responsible.

I appreciate the effort involved in the thorough characterization of the phenotypes in clone 16. However, there is a real possibility that the observed effects are at least partially due to something other than POLE1 that was fixed during the bottleneck process. The results for clone 16 would be significantly strengthened by confirmation with an additional clone. The results for clone 15 in Fig 1A and S1E are promising, but insufficient. High variability in clonal response would point more towards off target explanations.

Calling the effects of adding deltaCatalytic "partial rescue" seems a little bit of a stretch. The effects of synthesis in Fig. 5 are very modest.

The claim that processive synthesis has been restored by C-terminal POLE addition lacks sufficient supporting evidence. There is some degree of synthesis but it is small and, as the authors discuss, likely due to POLD synthesis at origins consistent with the model proposed in yeast.

The iPond results seem to argue against the idea that small amounts of residual POLE recycle to multiple origins and it isn't clear why this is argued in the discussion. One would expect at least some POLE to be captured during this process. The simpler explanation is that the high auxin degradation efficiency is indicating capture of only POLA/POLD synthesis at origins, consistent with what is seen in yeast.

Minor Comments

The effects of aux on POLE3 and POLE4 should be mentioned as well. The numbers in the table provided are variable, likely due to the small number of peptides. But what is seen seems to agree with POLE2 and is worth commenting on.

Abstract Ln 11: "falling behind" does not seem correct. More like "less is known about POLE1 roles in

replication initiation in human cells". And this is due to the recent advances in genetic manipulation that the authors take advantage of in the current study.

Pg 1 Ln 44: "...after the initial bypass of polymerases past one another."

The introduction and other sections have very long single paragraphs which are comprised of different separate thoughts. They should be set apart as separate paragraphs.

Pg 2 Ln 127: (Fig. 1A)

Pg 2 Ln 140: This reference does not appear in the citations.

Fig. 1 legend, next to last line: "...based off of..."

Pg 9 Ln 496: "In the study..."

Pg 11 Ln 562" "...epsilon is conserved..."

Reviewer #2 (Remarks to the Author):

The study by Vipat and co-workers investigates the role of DNA polymerase epsilon (POLE) in transformed human tissue culture cells. The team generated two independent clones that harbor an auxin-inducible POLE1 degron system in which the F-box protein Os TIR1 is under the control of doxocycline. Combined treatment with auxin/doxocycline (aux/dix) yields to a significant loss of POLE1, although residual protein is still visible on western blots and appears also to be present in one of the three iPOND mass spectrometry experiments. The quality of the degradation is also dependent on the passage number of the cell lines, as later passages accumulate DNA methylation of the doxocycline-inducible promoter that renders it unresponsive to the drug. POLE1 is required for DNA synthesis, and in yeast, Pol2 (its counterpart) has been implicated in the co-dependent recruitment of CDC45 and GINS to the MCM2-7 core helicase to form the CDC45:MCM2-7:GINS (CMG) complex. Here the authors provide evidence that the degradation of POLE1 in human cells does not impact CMG formation and activation. Although this is potentially interesting, the data are still too preliminary to justify a strong conclusion. Some of the data is difficult to interpret, partly because it's never quite clear to what extent POLE1 has been removed. The effects on DNA synthesis are dramatic, and well documented with independent approaches, but the core question about CMG assembly and function is less rigorously investigated. In addition, some of the experiments are performed in a single cell clone of a single parental cell line (U2OS), making the findings highly specific to U2OS cells. The authors propose further that the system will allow them in the future to study the function of disease variants of POLE1. Based on the caveats listed below, this is not entirely convincing.

Concerns and suggestions:

Figure 1. Panel A shows western blots of two independent POLE1 degron clones (15 and 16). Both clones show residual protein expression. They also show that expression levels of the POLE1 degron are slightly lower than in the parental cell line. It would be helpful if all panels would show the results for both clones.

Figure 2. Panels A and B show western blots of soluble and chromatin fractions. In the absence of ATR inhibition, it is impossible to assess the loading status of MCM4. Also, it seems that ATR inhibition in clone 16 in the absence of aux/dox did not lead to a significant increase in phosphorylated MCM4, CDC45 and SLD5 (Figure 2B lane 3).

Figure 3. The fiber experiments in panel D are verbally described but not quantified. The text implies

that the “dots” in the right panel represent firing events at a similar frequency as on the left (if I understand correctly). Why was there no attempt to quantify DNA synthesis events? The iPOND data is difficult to reconcile with the mass spectrometry data provided in the supplement. For experiments 1 and 3, POLE1 seemed to be absent in the “doxaux” fractions, but not in experiment 2. Yet, experiment 2 has still been included into the analysis. A separate table that lists “protein abundance” based on recovered fragments for each of the three experiments would be helpful to the reader.

Figure 4. It is expected that MCM7 loading onto chromatin remains unaffected by POLE1 depletion. A similar analysis should be performed for CDC45 and GINS. This would directly address the point of whether POLE1 plays a role in CMG assembly. C shows a very nice SMLM image of EdU/PCNA/MCM (which subunit?) colocalization. It is unclear to me why PCNA spots have not been quantified. PCNA loading is dependent on CMG activation. Aphidicolin could be added to block forks from elongation in the absence of aux/dox treatment.

Figures 5/6. This data convincingly shows that the ZnF domain in the non-catalytic C-terminus weakly interacts with POLE2. It also shows that in the amount of EdU incorporation in the absence of full-length POLE1 is very low. Cells complemented with this fragment support DNA replication initiation and very slow synthesis (Fig. 6). Whether this synthesis is indeed “processive” is unknown. I’m not sure on what basis the authors use this term. Also, these conclusions are based on a single cell clone. The bottom panel in Fig. 6D shows that the nucleoside analog incorporation shows small gaps. None of the stained fibers shows continuous incorporation. I think that argues that DNA synthesis is likely not processive. Contrary to the authors’ statement, it should be possible to determine origin density (inter-origin-distance) from these experiments as long as DNA is counterstained.

Figure S2. These data can be further analyzed and quantified. It also looks to me as if the G1 population is larger under aux/dox conditions.

Figure S3. Panels D and E. It would be helpful to show the same proteins with the different normalization procedures.

Calculation of number of replication forks: page 9 lines 469, 470. “given that EdU number \sim # of forks \times fork speed, we can expect about 3.6x more replication forks in dox/aux treated 1.6+Dcat cells.” Replication fork speed can only be reliably determined by DNA combing when fibers are stretched out uniformly.

Reviewer #3 (Remarks to the Author):

In this manuscript, Vipat et al addressed the non-catalytic role of DNA polymerase E1 C terminus in replication initiation in human cells. Although this has been investigated in lower eukaryotes like yeasts, it is still an important one to look at in mammals. They adopted an inducible degron system to overcome the essentiality of this gene and performed in vivo biochemical, protein interaction and DNA replication assays. The main conclusion of this study is that Pole1-C is not required for CMG assembly in humans, unfortunately it needs more convincing supports.

Major issue:

1. The evidence is not sufficient to support POLE1 depletion does not affect CMG assembly. The best way to monitor that is GINS or CDC45 IP. If the authors can carefully compare CMG level (by GINS or CDC45 IP) in POLE1 degron cell line. Or compare the time-course chromatin-loaded Cdc45 and GINS in synchronized cells in Pole1 mutants. These are minimal requirements to support the model. In the case of lacking good antibodies, I’d like to recommend two ways to add a tag in previous published work to prevent tagging problems: N-ter tag of SLD5 (Villa et. al. 2021) or internal tag of CDC45 (Jones et. al. 2021).

2. Fig1, the author used CRISPR/Cas9 to knock-in (KI) oSTIR1 under a doxycycline-inducible promoter into the AAVS1 "safe harbor" locus, and added a mAID-mCherry tag at the C-terminus of endogenous POLE1. The migration is a bit confusing, POLE1 seems bigger in clone 15 than in clone 16, whereas in Figure S1G, POLE1 has two bands? The protein level of POLE1 in Clone 15 and 16 is less than that in U2OS. And Clone 16 indeed grows slower compared to wild-type U2OS in the absence of mAID induction. The band of POLE1 should be verified and the protein level should be quantified. It will be good to know how much POLE1 left by inducing degron compare with the physiological POLE1? It will also be good to show whether the mAID-mCherry tag affects POLE1 in humans.

3. Fig2A&B, Clone 16 and U2OS, after ATRi treatment, the protein bands of POLE are different with different levels as well. pMCM, CDC45 and SLD5 on chromatin are also quite different. Then, in Clone 16, ATRi vs dox/aux+ATRi, loss of POLE1 promoted pMCM when ATR is inhibited. These experiments should be repeated 3 times using different clones or biological repeats, and at least protein level of chromatin fraction should be quantified. The signal of sample dox/aux- ATRi- aph- might be used for normalization.

4. Fig3E, if DNA synthesis level is quite different between DMSO and dox/aux sample, how reliable IPOND method will be? In the same page, can the authors explain depletion of POLE1 cause checkpoint activation, but why RPA is less abundant in dox/aux sample? One explanation is depletion of POLE1 slow down CMG, but the authors need evidence to prove that.

5. Fig5B, because endogenous POLE1 still available, there is no evidence prove that POLE1 truncations are as good as endogenous FL, which makes experiment less reliable. In human replisome structure (Jones et. al. 2021), POLE binds CMG with its non-catalytic domain. Which means if this experiment, catalytic domain truncation shouldn't bind with POLE2 (it actually does) and replisome (but it binds MCM7). A better way to perform the experiment is expressing truncations in POLE1-degron cell-line.

6. Figure S4B, by blot, the author can see ubiquitylated MCM7 is much less than unmodified MCM7, probably because there is large amount of DH-MCM on chromatin, which means it's not a reliable way to detect disassembly/unloading events. Back to major point, P97 inhibition plus GINS (Villa et.al. 2021) or CDC45 IP will be a better way to detect CMG disassembly.

7. if POLE1 depletion wouldn't affect CMG assembly and activation but only DNA synthesis, then CMG will be protected by DNA fork structure to avoid being disassembled (Deegan et. al. 2021; Low et.al. 2020). Such concerns should be discussed as well.

We thank the reviewers for their constructive comments and suggestions that helped significantly improve our manuscript and strengthen our conclusions. Please see below our point-by-point response to the reviewers' comments: the reviewers' comments are in **bold**, our responses are in regular font, quotes from the manuscript are *italicized* and in quote marks (changes in the revised manuscript are in *red*):

Reviewer #1 (Remarks to the Author):

DNA polymerase epsilon is critical replication polymerase that is responsible for normal leading strand synthesis. Older studies from yeast showed that Pol epsilon was dispensable for normal cell growth while Pol delta, the primary lagging strand pol was essential. Newer results have placed Pol epsilon interacting with the CMG helicase and actually taking over leading strand synthesis after initiation via Pol alpha/Pol delta. Whether this model holds in mammalian cells has not been directly tested. The current study begins to address these issues by testing a number of replication and DNA synthesis outcomes using an auxin-dependent degron approach. They make several notable and interesting findings. Removing 90% of pol epsilon catalytic subunits dramatically impairs cell proliferation and DNA synthesis. However there are large numbers of focus-like DNA synthesis events by DNA fiber analyses. Adding the Zinc finger-containing C-terminal fragment can partially restore shorter DNA synthesis events, suggesting that the Pol epsilon Cterm-CMG interaction may stabilize the replication initiation enough to start more DNA synthesis. Overall this study will be of significance to the field since it confirms and extends the model of initial DNA synthesis proposed in yeast and demonstrates the utility of a new tool for polymerase protein reduction.

We thank the reviewer for recognizing the value of our study.

Major Comments

Based on the 72 hr growth curve and described passaging issues the C-terminal AID-mCherry is likely impairing POLE function even in the absence of auxin-induced degradation. It is also unclear what the role of the C-terminal mCherry is. If the deltaCatalytic-mCherry construct suppressed the EdU incorporation defect to the same extent as the deltaCatalytic construct, this would clarify what is responsible.

Initially, the role of mCherry was to help select the knock-in clones, quantify the efficiency of POLE1 depletion, possible fluorescence microscopy experiments, and so on. Unfortunately, the fluorescence signal from POLE1-mCherry is so low, that we can't easily detect it by fluorescence microscopy or flow cytometry. Long exposures with high intensity excitation did confirm that mCherry is expressed, but it could not be used for any meaningful experiments. We agree that mCherry could have a slight effect on the normal function of POLE1 in our cells based on the slower growth, and therefore we have now added additional data comparing untreated U2OS and two POLE1-mAID-mCherry clones showing no significant effect of the tagging on replication dynamics:

- 1) The level of POLE1 protein (quantified from western blot) – (Fig S1e).
- 2) The level of DNA synthesis/EdU incorporation (Fig. S1f).
- 3) The percent of cells in S-phase (Fig. S1g).
- 4) DNA fiber length (fork speed) (Fig. Sh-i).

Based on these data we decided against complementing the cell line with mCherry-tagged Δ cat.

We did not observe any issues cultivating these cell lines, the cells grow well and look normal even after months in culture, using POLE1-mAID-mCherry for DNA synthesis. Our issue described in Supplementary fig. 1 concerns the methylation of dox-inducible promoter needed for the expression of osTIR1 for subsequent POLE1 degradation. We believe that the main reason for promoter methylation in the absence of dox-aux, is the location of the knocked-in osTIR1 gene in the AAVS1 locus which is not normally expressed. For our future studies we are working to use a different locus for the osTIR1 knock-in.

I appreciate the effort involved in the thorough characterization of the phenotypes in clone 16. However, there is a real possibility that the observed effects are at least partially due to something other than POLE1 that was fixed during the bottleneck process. The results for clone 16 would be significantly strengthened by confirmation with an additional clone. The results for clone 15 in Fig 1A and S1E are promising, but insufficient. High variability in clonal response would point more towards off target explanations.

We agree with the reviewer. In order to use truly different and independently obtained clones, instead of clone 15, we now provide more data with clone 1.6 which was obtained in a separate knock-in experiment and has different antibiotic resistance. Clone 1.6 was initially used in Fig. 6 as a control for all the Δ cat experiments, and we now include additional data with this clone throughout the manuscript. The data are generally consistent with what we reported for clone 16. In addition to general characterization described above, the following experiments were added for clone 1.6:

- 1) POLE1 depletion in response to dox/aux – western blot (Fig. 1a)
- 2) Growth curves with and without dox/aux (Fig. 1b)
- 3) EdU/DNA flow cytometry (Fig S1j)
- 4) Replication initiation in response to ATRi – proteins on NIC, and EdU incorporation (Fig. S2a-c)
- 5) CMG assembly on chromatin after synchronization (Fig.2c, S2c-d)
- 6) Long EdU treatment/EdU incorporation quantification (Fig. 3c)
- 7) DNA fiber analysis (Fig. S3c-d)

Additionally, we reproduced the key experiments for the Δ cat part of the manuscript in an additional Δ cat clone (Δ cat.25):

- 1) Western blot characterization (Fig. S6a)
- 2) EdU incorporation speed (Fig. S6b)
- 3) DNA fiber analyses (Fig. S6 e-f)

Calling the effects of adding deltaCatalytic “partial rescue” seems a little bit of a stretch. The effects of synthesis in Fig. 5 are very modest.

We are convinced that there is a significant difference between the replication dynamics by POLE1-depleted cells and cells missing only the catalytic domain of POLE1. The fiber patterns are strikingly different. We have now added a fiber experiment with longer IdU pulse for POLE1-depleted clones (Fig. S3d), and still could not observe any significant tracks easily seen in Δ cat samples, the dotted phenotype remains. We agree that quantitatively the DNA synthesis changed only about 3-4 fold, but we believe that the fact that Δ cat allows DNA replication to proceed past the initial steps can be called a partial rescue.

The claim that processive synthesis has been restored by C-terminal POLE addition lacks sufficient supporting evidence. There is some degree of synthesis but it is small and, as the authors discuss, likely due to POLD synthesis at origins consistent with the model proposed in yeast.

We agree that we have not directly tested the processivity of the DNA synthesis, and we therefore removed the claims of processive synthesis, substituting it with “continuous”. However, based on the length of the fibers we observe after prolonged incubation with the nucleotide analogs, DNA synthesis (likely by POLD) goes significantly beyond the initial synthesis at origins in the presence of Δ cat. According to our model, the initial synthesis by POLD can be observed in POLE-depleted cells as a “dotty” pattern in the fiber experiments. In the presence of Δ cat, DNA synthesis (likely by POLD) produces measurable tracks that progressively increase in length at least during the 1h that we tested in our fiber experiments.

The iPond results seem to argue against the idea that small amounts of residual POLE recycle to multiple origins and it isn't clear why this is argued in the discussion. One would expect at least some POLE to be captured during this process. The simpler explanation is that the high auxin degradation efficiency is indicating capture of only POLA/POLD synthesis at origins, consistent with what is seen in yeast.

We agree that POLA/POLD synthesis at origins is the most likely explanation, and it is stated in the manuscript.

Lines 356-358:

*“We propose that the DNA synthesis that we observe in the absence of POLE1 is the initial **POLA/POLD**-dependent synthesis that fails at the polymerase switch step.”*

Quick recycling and unstable association of leftover POLE1 with CMG during the initial stages of origin firing is one of the possible models that we wanted to discuss, and it would be consistent with the absence of POLE1 in the iPOND. Lines 348-352:

*“One possible explanation of these data is that the scarce POLE transiently associates with pre-RCs, ensuring replication initiation, and quickly dissociates, moving on to the next pre-RC. **This would agree with the absence of POLE1 in the iPOND pulldowns, but it** would contradict a recent study³⁶ showing that yeast DNA polymerase epsilon has a very low exchange rate at the replication fork in vitro and this rate only goes down with the decrease of the concentration of polymerase epsilon.”*

Minor Comments

The effects of aux on POLE3 and POLE4 should be mentioned as well. The numbers in the table provided are variable, likely due to the small number of peptides. But what is seen seems to agree with POLE2 and is worth commenting on.

We agree, but due to extremely low signals, even zero for some repeats (POLE4 has a non-zero signal only in one sample out of 6), we could not reliably include these data in the paper. We have added the following sentence (lines 222-223):

*“**POLE3 and POLE4 subunits' signals were very weak, and we therefore did not include them in the quantifications.**”*

Abstract Ln 11: “falling behind” does not seem correct. More like “less is known about POLE1 roles in replication initiation in human cells”. And this is due to the recent advances in genetic manipulation that the authors take advantage of in the current study.

We agree and this has been corrected – line 24:

“Less is known about POLE1 functions in DNA replication initiation in human cells.”

Pg 1 Ln 44: “...after the initial bypass of polymerases past one another.”

The introduction and other sections have very long single paragraphs which are comprised of different separate thoughts. They should be set apart as separate paragraphs.

Pg 2 Ln 127: (Fig. 1A)

Pg 2 Ln 140: This reference does not appear in the citations.

Fig. 1 legend, next to last line: “...based off of...”

Pg 9 Ln 496: “In the study...”

Pg 11 Ln 562” “...epsilon is conserved...”

We really appreciate reviewer’s help in editing the text, all the corrections have been included.

Reviewer #2 (Remarks to the Author):

<...> POLE1 is required for DNA synthesis, and in yeast, Pol2 (its counterpart) has been implicated in the co-dependent recruitment of CDC45 and GINS to the MCM2-7 core helicase to form the CDC45:MCM2-7:GINS (CMG) complex. Here the authors provide evidence that the degradation of POLE1 in human cells does not impact CMG formation and activation. Although this is potentially interesting, the data are still too preliminary to justify a strong conclusion. Some of the data is difficult to interpret, partly because it’s never quite clear to what extent POLE1 has been removed. The effects on DNA synthesis are dramatic, and well documented with independent approaches, but the core question about CMG assembly and function is less rigorously investigated.

We thank the reviewer for finding our study interesting. To strengthen our conclusion about POLE1-independent CMG assembly, we have now added the ATRi-based data for an additional clone (Fig.S2a-c), and a synchronization experiment showing CDC45 and SLD5 recruitment to chromatin during S-phase entry (Fig. 2c, S2c-d).

In addition, some of the experiments are performed in a single cell clone of a single parental cell line (U2OS), making the findings highly specific to U2OS cells.

We agree that one clone is not enough for any strong conclusions, and have now repeated the key experiments with the additional clone. Please see the list below.

Unfortunately, given the time constraints of the revision we could not reproduce the study in a different cell line, but we now specify in the abstract that our results are valid for the U2OS cell line.

The authors propose further that the system will allow them in the future to study the function of disease variants of POLE1. Based on the caveats listed below, this is not entirely convincing.

We agree that the current setup is not optimal for the further studies of the disease variants, and we are working to make a more stable system. We therefore removed this sentence from the discussion.

Concerns and suggestions:

Figure 1. Panel A shows western blots of two independent POLE1 degron clones (15 and 16). Both clones show residual protein expression. They also show that expression levels of the POLE1 degron are slightly lower than in the parental cell line. It would be helpful if all panels would show the results for both clones.

We agree that characterization of at least two clones is necessary and we have now added additional validation for all the critical experiments using independently obtained clone 1.6:

- 1) POLE1 depletion in response to dox/aux – western blot (Fig. 1a)
- 2) Growth curves with and without dox/aux (Fig. 1b)
- 3) EdU/DNA flow cytometry (Fig S1j)
- 4) Replication initiation in response to ATRi – proteins on NIC, and EdU incorporation (Fig. S2a-c)
- 5) CMG assembly on chromatin after synchronization (Fig.2c, S2c-d)
- 6) Long EdU treatment/EdU incorporation quantification (Fig. 3c)
- 7) DNA fiber analysis (Fig. S3c-d)

Additionally, we reproduced the key experiments for the Δ cat part of the manuscript in an additional Δ cat clone (Δ cat.25):

- 1) Western blot characterization (Fig. S6a)
- 2) EdU incorporation speed (Fig. S6b)
- 3) DNA fiber analyses (Fig. S6 e-f)

The levels of POLE1 in U2OS and the used clones were quantified by western blot (summarized in fig. S1e) and we do not see a significant difference in the expression levels.

Figure 2. Panels A and B show western blots of soluble and chromatin fractions. In the absence of ATR inhibition, it is impossible to assess the loading status of MCM4. Also, it seems that ATR inhibition in clone 16 in the absence of aux/dox did not lead to a significant increase in phosphorylated MCM4, CDC45 and SLD5 (Figure 2B lane 3).

We have repeated this experiment several times, and present a better version in the revised manuscript. Additionally, we include this experiment with clone 1.6 for independent validation (Fig. S2a-b). We agree that this type of experiment is not a good way to assess MCM loading in the absence of ATRi, as we are looking only at the nuclease-insoluble chromatin fraction. This experiment was used to look at the recruitment of CDC45 and SLD5, as well as phosphorylation of MCM4, associated with origin firing. To

assess MCM loading on chromatin we used chromatin flow cytometry (Fig. 4a-b) and are now additionally including the experiment assessing the levels of CMG components on chromatin during S-phase entry after synchronization (Fig. 2c, S2c-d).

Figure 3. The fiber experiments in panel D are verbally described but not quantified. The text implies that the “dots” in the right panel represent firing events at a similar frequency as on the left (if I understand correctly). Why was there no attempt to quantify DNA synthesis events?

We did not mean to imply that we can quantify the firing frequency in any way. Unfortunately, most “dots” are very dim and we can’t properly separate and analyze them. Additionally, “dottiness” is a common artifact of DNA fiber and DNA combing experiments (see, for example, reference images in a recent STAR protocol for DNA combing – Moore et al., 2022), so quantitative analysis of purely “dotty” patterns can’t be reliable. To further characterize the phenotype, we have now included the two-color fiber experiment for an additional clone (Fig. S2e), and the fiber experiments with 60-min pulses of IdU for both clones (Fig. S3d), that show similar dotty patterns, and no measurable tracks.

Lines 205-209:

“Since “dottiness” is a common artifact of DNA fiber and DNA combing experiments, quantitative analysis of purely “dotty” patterns can’t be reliable. To test if the “dots” represent extremely slow replication forks, we tried labelling ongoing DNA synthesis in POLE1-depleted cells for 60 min (Supplementary Fig. 3d), however, the phenotype did not change and we did not observe any tracks that could indicate continuous DNA synthesis.”

The iPOND data is difficult to reconcile with the mass spectrometry data provided in the supplement. For experiments 1 and 3, POLE1 seemed to be absent in the “doxaux” fractions, but not in experiment 2. Yet, experiment 2 has still been included into the analysis. A separate table that lists “protein abundance” based on recovered fragments for each of the three experiments would be helpful to the reader.

Indeed, in the experiment 2 we picked up a very small fraction of the leftover POLE1, and it is reflected on the plots (Fig. 3f-g), however, the depletion is still very strong, and we do not see any striking differences between repeat 2 and repeats 1 and 3. We included all three experiments on the plot clearly marked with repeat numbers, so all the data are available for interpretation.

We have added the requested table (Supplementary table 2), although due to varying efficiency of mass spec detection for different proteins, these data do not reliably represent relative protein abundance. Only changes in a specific protein can be properly assessed in our experiment.

Figure 4. It is expected that MCM7 loading onto chromatin remains unaffected by POLE1 depletion. A similar analysis should be performed for CDC45 and GINS. This would directly address the point of whether POLE1 plays a role in CMG assembly.

We agree that chromatin FACS of the other CMG components would strengthen our conclusions, but unfortunately, we have not been able to obtain any antibodies against GINS subunits or CDC45 that would work for chromatin flow cytometry experiments. CDC45 and GINS recruitment to the nuclease insoluble chromatin in POLE1-depleted cells in response to ATR inhibition is shown by western blot on figures 2 and S2. In order to strengthen our conclusions, we have now added an experiment showing that the

recruitment of CDC45 and SLD5 to chromatin during S-phase entry after synchronization (Fig. 2c, S2c-d) by western blot is not blocked by POLE1 depletion.

Lines 164-173:

“To further investigate CMG helicase assembly in POLE1-depleted cells, we synchronized U2OS, clone 16, and clone 1.6 cells using thymidine-nocodazole block, and added dox/aux to all the cells during the last 8 h of nocodazole treatment (Fig. 2c). Cells were collected 0, 3, 9 and 12 h following the release from nocodazole into dox/aux containing medium, and the levels of MCM, CDC45, and SLD5 on chromatin were analyzed. All three cell lines have mostly completed mitosis by 3 h, and U2OS cells were in S-phase during the 9 h and 12 h timepoints (Supplementary Fig. 2c). POLE1-depleted clones 16 and 1.6, as expected, could not significantly progress through S-phase due to the lack of efficient DNA synthesis, however, these cells were capable of loading MCM on chromatin in G1 and recruiting CDC45 and GINS to chromatin 9 - 12 h after the release from nocodazole (Fig. 2c). These data support our hypothesis that POLE1 is not required for CMG assembly in U2OS cells.”

C shows a very nice SMLM image of EdU/PCNA/MCM (which subunit?) colocalization. It is unclear to me why PCNA spots have not been quantified. PCNA loading is dependent on CMG activation. Aphidicolin could be added to block forks from elongation in the absence of aux/dox treatment.

An antibody against MCM6 was used for SMLM experiments (corrected in the methods section).

We thank the reviewer for the suggestion. We have now added the PCNA quantification by SMLM from the same experiments (Fig.4.f-g, S4e). Lines 248-252:

“Interestingly, SMLM analysis of PCNA foci in POLE1-depleted cells showed that both PCNA cluster density (Fig. 4f) and the amount of PCNA per focus (Fig. 4g) slightly decreased in response to POLE1 depletion, while the overall PCNA signal remained unchanged (Supplementary Fig. 4e). Lower number of PCNA molecules per fork is consistent with failure to establish lagging strand synthesis that normally harbors the majority of PCNA molecules.”

Figures 5/6. This data convincingly shows that the ZnF domain in the non-catalytic C-terminus weakly interacts with POLE2. It also shows that in the amount of EdU incorporation in the absence of full-length POLE1 is very low. Cells complemented with this fragment support DNA replication initiation and very slow synthesis (Fig. 6). Whether this synthesis is indeed “processive” is unknown. I’m not sure on what basis the authors use this term. Also, these conclusions are based on a single cell clone. The bottom panel in Fig. 6D shows that the nucleoside analog incorporation shows small gaps. None of the stained fibers shows continuous incorporation. I think that argues that DNA synthesis is likely not processive. Contrary to the authors’ statement, it should be possible to determine origin density (inter-origin-distance) from these experiments as long as DNA is counterstained.

We agree that our experiments can’t definitively prove that the DNA synthesis is processive, and have now removed this word from the text. However, dots and small gaps are very common artifacts of DNA fiber analysis (see, for example, reference images in a recent STAR protocol for DNA combing – Moore et al., 2022), so we can’t use the dottiness to argue for or against the processive synthesis.

Inter-origin distances are generally determined by DNA combing, where the fibers are uniformly stretched. In our fiber experiments, the fibers are often bent and/or broken, and it is rarely possible to

get a reliable measurement of an inter-origin distance for the POLE1-proficient cells, where these distances are quite long.

Figure S2. These data can be further analyzed and quantified. It also looks to me as if the G1 population is larger under aux/dox conditions.

Figure S2 in the original submission showed an experiment similar to the experiment shown on the panel 2c (the effect of ATRi on EdU incorporation), just with a two-color staining (EdU and DNA). We chose to quantify the one-color experiment (Fig. 2d-e) because of the partial bleedthrough in the two-channel experiment which makes the measurement of EdU incorporation less precise: while it shows the cell cycle outlook correctly, the exact EdU incorporation signal may be distorted. We have now completely removed the former figure S2, and instead added a one-color experiment for the additional clone 1.6 (Fig. S2c) that also shows an increase in EdU incorporation by POLE1-depleted cells in response to ATRi.

The cell cycle distribution in the absence of ATRi has been quantified on Fig. 1e-f, and given a very short treatment with ATRi (30-60 min), we did not expect any noticeable effects on the cell cycle. The absence of ATRi effect on the cell cycle is also evident from the one color EdU FACS on Fig. 2d.

Just like the reviewer mentioned, we also initially thought that G1 phase appears larger in dox/aux treated samples. However, the quantification performed after proper gating of G1 phase that included the limit on the DNA content (Fig. 1f), did not lead to this conclusion. This is additionally confirmed by the experiments with long EdU incubations (Fig. 3a), where cells are seen moving from G1 to S (starting to incorporate EdU) between the timepoints, confirming no G1 arrest.

Figure S3. Panels D and E. It would be helpful to show the same proteins with the different normalization procedures.

We have now added the missing panel – new panel 3f, and both normalizations are showing the same proteins.

Calculation of number of replication forks: page 9 lines 469, 470. “given that EdU number ~# of forks x fork speed, we can expect about 3.6x more replication forks in dox/aux treated 1.6+Dcat cells.” Replication fork speed can only be reliably determined by DNA combing when fibers are stretched out uniformly.

We agree that we can't reliably measure the absolute speed of replication forks, and we only estimate it based on the fiber analysis. However, the calculation mentioned by the reviewer takes into account only the relative changes in the replication fork speed, and not the absolute speed itself. Assuming that the fiber length measurements are indicative of the average fork speed, they allow us to estimate the fold change in the replication fork speed after treatment/POLE depletion. And, considering the fold change in overall DNA synthesis measured by the EdU incorporation experiments, we can estimate the fold change in the number of active forks.

Reviewer #3 (Remarks to the Author):

In this manuscript, Vipat et al addressed the non-catalytic role of DNA polymerase E1 C terminus in replication initiation in human cells. Although this has been investigated in lower eukaryotes like yeasts,

it is still an important one to look at in mammals. They adopted an inducible degron system to overcome the essentiality of this gene and performed *in vivo* biochemical, protein interaction and DNA replication assays. The main conclusion of this study is that POLE1-C is not required for CMG assembly in humans, unfortunately it needs more convincing supports.

We thank the reviewer for recognizing the importance of our work, and believe that with the additional data included in this resubmission, our conclusions are now strong enough for publication.

Major issue:

1. The evidence is not sufficient to support POLE1 depletion does not affect CMG assembly. The best way to monitor that is GINS or CDC45 IP. If the authors can carefully compare CMG level (by GINS or CDC45 IP) in POLE1 degron cell line. Or compare the time-course chromatin-loaded Cdc45 and GINS in synchronized cells in POLE1 mutants. These are minimal requirement to support the model. In the case of lacking good antibodies, I'd like to recommend two ways to add a tag in previous published work to prevent tagging problems: N-ter tag of SLD5 (Villa et. al. 2021) or internal tag of CDC45 (Jones et. al. 2021).

Unfortunately, due to the time constraints we were unable to establish either of the cutting-edge CMG purification techniques suggested by the reviewer – one of them required creating a knock-in cell line, and the other used overexpression of all CMG components together. Overexpressing just one N-terminally tagged SLD5 subunit proved to be inefficient for CMG pulldown in our hands/cells.

Therefore, as suggested by the reviewer, we performed a synchronization experiment showing CDC45 and SLD5 loading on chromatin as cells enter S-phase, for two POLE1-mAID-mCherry clones (Fig. 2c, S2c-d).

Lines 164-173:

“To further investigate CMG helicase assembly in POLE1-depleted cells, we synchronized U2OS, clone 16, and clone 1.6 cells using thymidine-nocodazole block, and added dox/aux to all the cells during the last 8 h of nocodazole treatment (Fig. 2c). Cells were collected 0, 3, 9 and 12 h following the release from nocodazole into dox/aux containing medium, and the levels of MCM, CDC45, and SLD5 on chromatin were analyzed. All three cell lines have mostly completed mitosis by 3 h, and U2OS cells were in S-phase during the 9 h and 12 h timepoints (Supplementary Fig. 2c). POLE1-depleted clones 16 and 1.6, as expected, could not significantly progress through S-phase due to the lack of efficient DNA synthesis, however, these cells were capable of loading MCM on chromatin in G1 and recruiting CDC45 and GINS to chromatin 9 - 12 h after the release from nocodazole (Fig. 2c). These data support our hypothesis that POLE1 is not required for CMG assembly in U2OS cells.”

We have also repeated the ATRi-based experiments, and are now including improved versions using two mAID clones and the earliest available passages to minimize the leftover POLE1 (Fig. 2a-b, Fig.S2a-b).

We can't think of an alternative explanation for GINS and CDC45 recruitment to chromatin during ATRi-induced origin firing or S-phase entry, and have to conclude that POLE1-depleted U2OS cells can assemble CMG.

2. Fig1, the author used CRISPR/Cas9 to knock-in (KI) osTIR1 under a doxycycline-inducible promoter into the AAVS1 “safe harbor” locus, and added a mAID-mCherry tag at the C-terminus of endogenous POLE1. The migration is a bit confusing, POLE1 seems bigger in clone 15 than in clone 16, whereas in Figure S1G, POLE1 has two bands?

The seemingly slow migration is due to the slight bend in the low-percentage acrylamide gel. All the clones used in the study show identical POLE1-mAID-mCherry migration, that is slightly slower than the endogenous untagged protein, as expected. We have now re-run this experiment with clones 16 and 1.6 (Fig.1a).

The second (upper) band of POLE1 on former panel S1G (now S1I) represents the protein that got stuck between the stacking and resolving gels. As the protein is quite large (261kDa), this happens sometimes. For our semi-quantitative use of the western blot technique this artifact does not affect the main messages of the panels. Uncropped blots are provided for all panels with the resubmission.

We do see a second (lower) band in POLE1 blots in some experiments (Fig.2, S2), but as this band is present in U2OS as well as in the mAID clones, and depleted by dox/aux treatment, we conclude that it is an isoform of POLE1.

The protein level of POLE1 in Clone 15 and 16 is less than that in U2OS. And Clone 16 indeed grows slower compared to wild-type U2OS in the absence of mAID induction. The band of POLE1 should be verified and the protein level should be quantified. It will be good to know how much POLE1 left by inducing degron compare with the physiological POLE1? It will also be good to show whether the mAID-mCherry tag affects POLE1 in humans.

We agree that mCherry could have a slight effect on the normal function of POLE1 in our cells, and therefore we have now added additional data comparing untreated U2OS and two POLE1-mAID-mCherry clones showing no significant effect of the tagging on replication dynamics:

- 1) The level of POLE1 protein (quantified from western blot) – (Fig. S1e).
- 2) The level of DNA synthesis/EdU incorporation (Fig. S1f).
- 3) The percent of cells in S-phase (Fig. S1g).
- 4) DNA fiber length (fork speed) (Fig. Sh-i).

Unfortunately, it is impossible to assess the level of leftover POLE1 in POLE1-depleted cells, because of the presence of the dox-resistant subpopulation in every experiment. We have attempted a flow cytometry experiment to quantify the leftover POLE1, but the antibodies we tested did not work for flow cytometry, and the mCherry signal is too weak for microscopy or flow cytometry. The absence of POLE in the iPOND pulldowns suggests that the POLE1 depletion is highly efficient (Fig. 3f-g).

Regarding verification of the POLE1 band, the standard approach would be a knockout or a knockdown. We believe that our mAID-induced knockdown in two independently obtained clones repeatedly showing drastic decreases of POLE1 (Fig. 1a, 2a-b, S2a-b) could be considered an additional validation for the used antibody/band.

3. Fid2A&B, Clone 16 and U2OS, after ATRi treatment, the protein bands of POLE are different with different levels as well. pMCM, CDC45 and SLD5 on chromatin are also quite different. Then, in Clone 16, ATRi vs dox/aux+ATRi, loss of POLE1 promoted pMCM when ATR is inhibited. These experiments

should be repeated 3 times using different clones or biological repeats, and at least protein level of chromatin fraction should be quantified. The signal of sample dox/aux- ATRi- aph- might be used for normalization.

These experiments have been repeated multiple times on both clones, using the earliest available passages to minimize the dox-resistant population, and in this revised manuscript we have replaced the main panel with a better version (Fig. 2a-b), and added another panel with a similar experiment in clone 1.6 (Fig. S2a-b).

We consistently see stronger MCM4 phosphorylation, CDC45, and SLD5 signals in the nuclease insoluble fraction of POLE1-depleted cells (Fig. 2b, S2a) both with and without ATRi, it is especially evident with these samples with low dox-resistant population. CDC45/GINS recruitment to the nuclease-insoluble chromatin of POLE1-depleted cells during ATRi-induced origin firing strongly supports the hypothesis of POLE1-independent CMG assembly.

Since western blot is widely considered to be a semi-quantitative method, and the relative increases in pMCM, CDC45 and SLD5 were largely dependent on the passage number of the clone, re-using the dilution of the primary antibody, and other factors, the quantification of these blots could not be reliably performed: we see an increase every time, but the specific value of the increase differs too much, giving a very broad distribution with no statistically significant difference.

We decided not to include these quantifications in the manuscript. We believe that the new versions of the western blot panels are informative and representative.

4. Fig3E, if DNA synthesis level is quite different between DMSO and dox/aux sample, how reliable IPOND method will be? In the same page, can the authors explain depletion of POLE1 cause checkpoint activation, but why RPA is less abundant in dox/aux sample? One explanation is depletion of POLE1 slow down CMG, but the authors need evidence to prove that.

We agree that the reliability of the iPOND is limited in this case. However, to circumvent this, we used two different normalization approaches, and three independent experimental repeats. Additionally, we do not rely on the iPOND for any of our conclusions except in the case of the absence/presence of DNA polymerases at the DNA synthesis sites, which we can confidently conclude from the three experimental repeats.

We agree that the lack of increase in the RPA levels in the dox/aux-treated sample is unexpected. However, there is a large variability between the RPA subunits and between three repeats, so the

decrease can't be considered reliable either. We do see a strong ssDNA induction and ATR activation (Fig. 2a, 2f, S2b, S2f) in dox/aux-treated samples, so we can't argue that CMG is stopped, and indeed we believe that CMG is active at least for some time after replication fails (as stated in the discussion). Having CMG at a significant distance from the failed DNA synthesis could explain the decrease of MCM relative to the histones in the iPOND – MCM could be far enough not to be pulled down with the newly synthesized DNA. If most of the extra RPA is also near the runaway CMG and further from the short nascent DNA fragments - for example, if the separated DNA strands could re-anneal behind the CMG in the absence of DNA synthesis - this could explain the lack of RPA increase in the iPOND.

However, this is just a speculation, and, given the limited reliability of the iPOND, we can't make any conclusions about the RPA location at this point. DNA unwinding is discussed in the manuscript (lines 366-370):

“Additionally, based on the accumulation of single-stranded DNA and ATR activation in POLE1-depleted cells (Fig. 2a, f), it is likely that MCM continues DNA unwinding for some time after DNA synthesis fails. It remains unclear what happens to these failed replication complexes, but being located far enough from the failed DNA synthesis could explain their decrease in the iPOND pulldown.”

5. Fig5B, because endogenous POLE1 still available, there is no evidence prove that POLE1 truncations are as good as endogenous FL, which makes experiment less reliable. In human replisome structure (Jones et. al. 2021), POLE binds CMG with its non-catalytic domain. Which means if this experiment, catalytic domain truncation shouldn't bind with POLE2 (it actually does) and replisome (but it binds MCM7). A better way to perform the experiment is expressing truncations in POLE1-degroun cell-line.

We chose to perform this experiment in 293FT cells due to higher transfection efficiency in this cell line. The goal of the experiment was to confirm that the truncation mutants can maintain the interactions with POLE1 and CMG as we expected from the literature.

The catalytic domain alone does not bind POLE2 in our experiment, in agreement with all available structures.

We conclude that C-terminal Zn-finger region of POLE1 mediates POLE1-POLE2 interaction in human cells, as it was first shown by Baranovskiy et al., 2017.

While it is indeed well known that the C-terminal non-catalytic domain of POLE1 binds CMG, the exact position of the N-terminal catalytic domain of POLE1 relative to CMG has not been well determined, and it is absent in the structure mentioned by the reviewer. Based on the low-resolution model from yeast, its position changes when the POLE switches from the inactive conformation during the replication initiation to the active form after the initial DNA synthesis by POLD (Zhou et al., 2017). Yeast N-terminal domain of Pol2 has been

shown to participate in CMG assembly (Meng et al., 2020), implying a possible interaction with CMG, probably in the inactive conformation. This is in agreement with our data showing that both N-and C-terminal domains of POLE1 can co-precipitate with MCM. It is also discussed in the manuscript (lines 415-420):

“We found that both N-terminal catalytic domain and C-terminal non-catalytic domain of POLE1 can co-precipitate with MCM (Fig. 5b). According to structural studies of yeast⁴⁴ and human⁴³ replisomes, the active conformation of the PolE is connected to CMG solely through the C-terminal part of Pol2. However, the N-terminal domain of Pol2 participated in CMG binding in the inactive conformation⁵. In agreement with these findings, the N-terminal domain of Pol2 has recently been shown to play a role in mediating CMG-PolE interaction in yeast⁴⁵. Our data suggest a possibility for a similar mechanism in human cells.”

With regards to truncation mutants being as good as endogenous full-length protein, figure 5b indeed can't be used to address this question. However, based on the iPOND data (Fig. 6f), Myc-tagged POLE1 CTD associated with nascent DNA even in the presence of the endogenous protein (lane 5), which implies that its protein-protein interactions allow it to be recruited to the replisome, and it can compete with the endogenous FL protein.

6. Figure S4B, by blot, the author can see ubiquitylated MCM7 is much less than unmodified MCM7, probably because there is large amount of DH-MCM on chromatin, which means it's not a reliable way to detect disassembly/unloading events. Back to major point, P97 inhibition plus GINS (Villa et.al. 2021) or CDC45 IP will be a better way to detect CMG disassembly.

This line of inquiry was prompted by the iPOND result showing MCM decrease relative to DNA. But, as mentioned above, having CMG at a significant distance from the failed DNA synthesis could explain the decrease of MCM relative to the histones in the iPOND – MCM could be far enough not to be pulled down with the newly synthesized DNA.

We agree that looking at the ubiquitylated MCM7 on chromatin without immunoprecipitation is not a reliable way to look at CMG disassembly, and we have removed this panel from our manuscript. Based on the non-decreasing level of MCM on chromatin even after 16h of continuous origin firing after POLE1 depletion, and the point that the reviewer is making below, we believe that MCM is not unloaded when replication fails in the absence of POLE1.

7. if POLE1 depletion wouldn't affect CMG assembly and activation but only DNA synthesis, then CMG will be protected by DNA fork structure to avoid being disassembled (Deegan et. al. 2021; Low et.al. 2020). Such concerns should be discussed as well.

We agree and have now added this to the discussion (lines 375-381).

“A recent series of publications^{31,39,40} demonstrated that the DNA strand excluded by MCM helicase masks the site on MCM required for its recognition by CUL2/LRR ubiquitin ligase, preventing the disassembly of CMG helicase. In our model, where replication can't be completed and MCM encircles one strand of DNA, allowing the excluded strand to prevent its disassembly, any unloading of the activated CMG from DNA is highly unlikely.”

REVIEWER COMMENTS

Reviewer #1 (Remarks to the Author):

The characterization of the second clone has significantly strengthened and improved the manuscript.

One very very tiny complaint is that the dashed lines in the Fig 1b legend are not really distinguishable from the unbroken lines.

Reviewer #3 (Remarks to the Author):

The revised manuscript has addressed the most of concerns except the one related to CMG. The authors did the chromatin fraction assay to prove POLe is dispensable for CMG assembly. It is fine to show chromatin-bound CDC45 and GINS. But first, the experiment should be repeated at least 3 times with quantification. Second the experiment should include other controls, mitotic samples for example can be a good control. Because this's the most important conclusion of this paper, which is clearly against current model. Such a claim should be supported by multiple very solid data. GINS or CDC45 IP is also highly recommended. In my opinion, the manuscript should only be considered with more evidence in this critical point.

We thank the reviewers for their additional comments and suggestions. Our point-by-point responses are included below.

Reviewer #1 (Remarks to the Author):

The characterization of the second clone has significantly strengthened and improved the manuscript.

One very very tiny complaint is that the dashed lines in the Fig 1b legend are not really distinguishable from the unbroken lines.

We are grateful to the reviewer for recognizing the improvements in our manuscript and pointing out the unclear legend. We have changed the legend to more clearly represent the lines.

Reviewer #3 (Remarks to the Author):

The revised manuscript has addressed the most of concerns except the one related to CMG. The authors did the chromatin fraction assay to prove POLe is dispensable for CMG assembly. It is fine to show chromatin-bound CDC45 and GINS. But first, the experiment should be repeated at least 3 times with quantification. Second the experiment should include other controls, mitotic samples for example can be a good control. Because this's the most important conclusion of this paper, which is clearly against current model. Such a claim should be supported by multiple very solid data. GINS or CDC45 IP is also highly recommended. In my opinion, the manuscript should only be considered with more evidence in this critical point.

As with any experiment that we included in the manuscript, the synchronization experiment was repeated over 3 times before the submission. We are now including additional two independent repeats of this experiment (supplementary figures 2f and 2g). Additionally, we are including two additional repeats of the ATRi/fractionation experiment (supplementary figures 2c and 2d). Unfortunately, the semi-quantitative nature of these experiments does not allow a reliable quantification, but we hope that showing three independent repeats for two independently obtained clones makes our data more convincing.

Mitotic samples were included in all synchronization experiments as „0h post nocodazole release” samples. We have amended the text to make it clear:

Line 166:

Cells were collected 0 (mitosis), 3, 9 and 12 h following the release from nocodazole into dox/aux containing medium, and the levels of MCM, CDC45, and SLD5 on chromatin were analyzed.